ecology, environmental science

rainforest, biodiversity, global change, biogeography, community assembly, occupancy

**Author for correspondence:**
Daniel Gorczynski
e-mail: daniel.gorczynski@rice.edu

# Tropical mammal functional diversity increases with productivity but decreases with anthropogenic disturbance

Daniel Gorczynski[1,2], Chia Hsieh[1,2], Jadelys Tonos Luciano[1,2], Jorge Ahumada[3], Santiago Espinosa[4,5], Steig Johnson[6], Francesco Rovero[7,8], Fernanda Santos[9], Mahandry Hugues Andrianarisoa[10], Johanna Hurtado Astaiza[11], Patrick A. Jansen[12,13], Charles Kayijamahe[14], Marcela Guimarães Moreira Lima[15], Julia Salvador[16] and Lydia Beaudrot[1,2]

[1]Department of Biosciences, Rice University, Houston, TX, USA
[2]Program in Ecology and Evolutionary Biology, Rice University, Houston, TX, USA
[3]Moore Center for Science, Conservation International, Arlington, VA, USA
[4]Facultad de Ciencias, Universidad Autónoma de San Luis Potosí, SLP, México
[5]Escuela de Ciencias Biológicas, Pontificia Universidad Católica del Ecuador, Quito, Ecuador
[6]Department of Anthropology and Archaeology, University of Calgary, Calgary, Alberta, Canada
[7]Department of Biology, University of Florence, Florence, Italy
[8]Tropical Biodiversity Section, MUSE-Museo delle Scienze, Trento, Italy
[9]Department of Mastozoology, Museu Paraense Emílio Goeldi, Belém, Pará, Brazil
[10]Centre ValBio, Ranomafana National Park, 312 Ifanadiana, Madagascar
[11]Organization for Tropical Studies, Puerto Viejo de Sarapiquí, Heredia, Costa Rica
[12]Smithsonian Tropical Research Institute, Balboa, Ancon, Republic of Panama
[13]Department of Environmental Sciences, Wageningen University, Wageningen, The Netherlands
[14]International Gorilla Conservation Programme, Kigali, Rwanda
[15]Biogeography of Conservation and Macroecology Laboratory, Institute of Biological Sciences, Universidade Federal do Pará, Pará, Brazil
[16]Wildlife Conservation Society, Mariana de Jesús E7-248 y Pradera, Quito, Ecuador

DG, 0000-0003-0395-0434; CH, 0000-0002-3715-4113; JTL, 0000-0001-8792-9717;
JA, 0000-0003-0953-9101; SE, 0000-0002-7416-7167; SJ, 0000-0003-2257-8949; FR, 0000-0001-6688-1494;
FS, 0000-0002-1886-446X; JHA, 0000-0001-7315-0892; PAJ, 0000-0002-4660-0314;
CK, 0000-0003-3726-095X; MGML, 0000-0002-2203-7598; JS, 0000-0002-8050-8765;
LB, 0000-0001-8135-7519

A variety of factors can affect the biodiversity of tropical mammal communities, but their relative importance and directionality remain uncertain. Previous global investigations of mammal functional diversity have relied on range maps instead of observational data to determine community composition. We test the effects of species pools, habitat heterogeneity, primary productivity and human disturbance on the functional diversity (dispersion and richness) of mammal communities using the largest standardized tropical forest camera trap monitoring system, the Tropical Ecology Assessment and Monitoring (TEAM) Network. We use occupancy values derived from the camera trap data to calculate occupancy-weighted functional diversity and use Bayesian generalized linear regression to determine the effects of multiple predictors. Mammal community functional dispersion increased with primary productivity, while functional richness decreased with human-induced local extinctions and was significantly lower in Madagascar than other tropical regions. The significant positive relationship between functional dispersion and productivity was evident only when functional dispersion was weighted by species' occupancies. Thus, observational data from standardized monitoring can reveal the drivers of mammal communities in ways that are not readily apparent from range map-based studies. The positive association between occupancy-weighted functional dispersion of tropical forest mammal communities and primary productivity

suggests that unique functional traits may be beneficial in more productive ecosystems and may allow species to persist at higher abundances.

## 1. Introduction

A broad range of factors can affect the structure of biodiversity in communities [1–4]. Comparative studies have shown that historical, environmental and anthropogenic factors can alter community taxonomic composition and abundances [5–7]. In addition to taxonomic approaches, the assembly of biological communities can also be assessed using trait-based approaches. Traits are relevant ecological characteristics of species, such as physiological attributes and resource acquisition strategies, that can provide a more mechanistic understanding of community responses to assembly processes [8]. Incorporating traits can improve on taxonomically oriented approaches because some deterministic assembly processes, such as environmental filtering and biotic interactions, act on traits rather than species' identities. Understanding the relative influences of historical, environmental and anthropogenic factors on communities of mammals is of particular interest because mammals perform unique and vital functions in their environment due to their large range in body sizes, dietary requirements and home range sizes [9]. Moreover, mammals are severely impacted by anthropogenic change due to these attributes [10], particularly in tropical ecosystems [11]. Here, we examine four potential drivers of tropical mammal community assembly.

First, the species pool hypothesis states that the number of species in a community is proportional to the number of species that occur in the larger region encompassing the community, or the species pool [12,13]. A variety of broad-scale ecological and evolutionary patterns influence species pools, including speciation, migration and extinction [14]. In this way, species pools capture differences in the regional evolutionary context from which realized communities form. In general, species pool size, both in number of species and number of traits, contributes to how much of the available niche space can be occupied [4,15,16]. Small species pool functional richness can result in communities with open niches while large species pool functional richness can allow for more complete filling of available niches.

Secondly, spatial environmental heterogeneity can increase an ecosystem's capacity for biodiversity [17,18]. The greater the environmental and habitat diversity, the greater variety of functional traits a system can contain [19]. Habitat heterogeneity is capable of promoting greater coexistence of species with a wide array of traits through species sorting [1,3,20]. Niche packing and specialization associated with high diversity of species in the tropics could be a result of high habitat heterogeneity and species sorting, with each species possessing functional traits that confer high fitness in a specific set of conditions [21].

Thirdly, primary productivity is an important correlate of plant and animal biodiversity across large spatial scales [22–24], potentially due to increased energetic capacity of the ecosystem and greater energy allocation to individual organisms [25]. Two hypothesized mechanisms of the diversity–productivity relationship are based on resource availability and niche diversity. Resource availability in high productivity

systems could benefit all species indiscriminately [23,26] resulting in consistent functional diversity regardless of productivity. Alternatively, productivity could favour species with unique functional traits, resulting in increased functional diversity in high productivity systems [25,27,28]. The relationship between community weighted functional diversity and productivity remains unexplored and could address the ecological mechanism of the species–productivity relationship by incorporating relative abundances of different species in the community [29].

Finally, disturbance can play an important role in community assembly and has been shown to both increase trait diversity in some systems and decrease trait diversity in others. Natural disturbance, such as fire, can create novel habitats that support new functional traits [19] and prevent competitive exclusion [30]. Anthropogenic disturbance, however, tends to negatively affect biodiversity through the selection of homogenized disturbance-adapted traits [31] resulting in local extinctions. In the tropics, increased anthropogenic disturbance has been linked to decreased mammal trait diversity [32,33]. Comparing protected areas exposed to different degrees of anthropogenic disturbance may reveal the relative importance of anthropogenic disturbance in tropical mammal community assembly in comparison to other drivers.

While species pools, habitat heterogeneity, primary productivity and anthropogenic disturbance are known drivers of community assembly, their relative importance and directionality in structuring mammal communities in the tropics remains uncertain. Previous studies have shown that tropical mammals have low functional dispersion (i.e. mean distance of individual species to the centroid of all species in a community along functional trait dimensions; [34]) in comparison with temperate communities [35], and that productivity is negatively associated with functional dispersion globally [36]. However, most previous investigations of global patterns of mammal functional diversity have not used observational data but have instead relied on range maps to determine community composition and calculate functional diversity [28,35–37, but see 38]. Range map-based analyses are unable to assess realized community composition or incorporate abundance data. Species abundances likely respond to local assembly processes and, therefore, may reveal aspects of community assembly that cannot be disentangled from range map data.

Here, we test for effects of species pools, habitat heterogeneity, primary productivity and anthropogenic disturbance on the functional diversity of global tropical mammal communities in 15 protected areas using standardized *in situ* camera trap data from the Tropical Ecology Assessment and Monitoring (TEAM) Network. We evaluate these four non-mutually exclusive hypotheses to assess their importance in determining functional diversity of mammals in tropical rainforest protected areas around the world. We predict that functional diversity will increase with species pool functional richness, habitat diversity and productivity, but decrease with human disturbance. We investigate functional diversity using two metrics—functional dispersion and functional richness—to address potential effects on functional trait abundances and functional trait composition, respectively. Lastly, we evaluate the contributions of occupancy as a proxy for abundance to functional diversity, by testing if and how occupancy affects associations between functional dispersion and environmental or anthropogenic variables.

## 2. Material and methods

### (a) Study sites

We examined mammal communities from 15 moist tropical forest protected areas around the world, including sites in Africa, Asia, Madagascar and the Neotropics (electronic supplementary material, table S1). All sites have been part of the TEAM Network and followed a standardized annual camera trapping protocol to monitor terrestrial (i.e. ground-dwelling) mammals [39]. We considered each site to be a community and were not able to account for variation within protected areas.

### (b) Data

We used species-specific protected area-level occupancy estimates derived from TEAM camera trap data from Beaudrot et al. [40] as a proxy for abundances. Occupancy is not equivalent to abundance [41] but was used because of its practical advantages for infrequently detected tropical mammal species [42]. Our use of occupancy-weighted functional diversity is particularly suitable for identifying underlying community assembly processes for two reasons. First, camera traps capture the realized local community, whereas range maps typically overestimate the geographical distribution of species by including potentially non-suitable habitat areas [43]. Secondly, by estimating imperfect detection, the occupancy values we used accounted for biases that result from imperfect detection and would otherwise skew results in favour of species that were more likely to be detected by camera traps. We used species occupancy values from the most recent year of TEAM data with occupancy estimates available, which varied among sites from 2012 to 2014. We included terrestrial mammal species with an average body mass greater than 1 kg that were monitored by the TEAM network camera traps. We did not consider strictly arboreal mammals, semi-aquatic mammals, volant mammals or mammals less than 1 kg, which could limit comparability to global studies that include additional mammals. The observed mammal communities ranged in species richness among protected areas from five to 31 species [38,40].

We collected functional trait data for all study species through a search of published literature and databases (table 1; electronic supplementary material, tables S2 and S3). We chose six functional traits to calculate functional diversity: (i) average body mass, (ii) diet composition, (iii) social or asocial behaviour, (iv) scansorial or entirely terrestrial substrate use, (v) activity period, and (vi) average litter size. These functional traits relate to both the response of the species to environmental conditions (response traits) and to the role of the species in an ecosystem (effect traits; [44]). Body mass affects the quality and quantity of resources necessary for survival, and also approximates the impact that the species may have on the ecosystem in terms of spatial range use, nutrient dispersal and trophic regulation. Diet composition characterizes the resources a species requires, but also identifies other taxa in the ecosystem with which a species potentially interacts. Social group size can alter species' allocation of time to different behaviours including foraging, predator avoidance and care of offspring, and also indicates how the species' impact will be distributed in space. Substrate use characterizes where a species can obtain resources and where a species will directly interact with an ecosystem. Activity period characterizes when a species obtains resources and interacts with an ecosystem. Finally, litter size characterizes the life-history strategy of a species and indicates how the ecological impact of a species can vary temporally based on population dynamics. As with any functional trait study, our results are in part dependent upon the traits considered and the diversity encompassed by functional metrics can change with the inclusion or exclusion of specific traits [45]. Traits used in this study were gathered at the species level and multiple traits were reduced to binary variables, which may reduce the breath of trait variation to some extent.

**Table 1.** Description of traits obtained for each species in the study. Values were drawn from published literature (see electronic supplementary material, table S3) and determined as the mean and/or consensus values from these published sources.

| functional traits and value descriptors |
| --- |
| ***body mass*** |
| calculated as average adult body mass |
| ***diet*** |
| binary values for five diet categories |
| *graze* |
| *browse* |
| *fruits/seed* |
| *invertebrates* |
| *vertebrates* |
| given a 1 if diet category composed >25% of average diet, 0 if diet category composed <25% of average diet |
| ***social*** |
| binary: given 1 if social, 0 if found solitarily or in pairs. |
| ***substrate use*** |
| binary: given 1 if able to climb, 0 if restricted to terrestrial surfaces |
| ***activity period*** |
| binary values for three activity periods |
| *diurnal* |
| *crepuscular* |
| *nocturnal* |
| given a 1 if known to be consistently active during the time period, 0 if known to not be consistently active during the time period |
| ***litter size*** |
| calculated as average number of offspring produced in a litter |

To preserve total inertia and distance between the same species occurring in different assemblages (e.g. [46]), we calculated functional dispersion [34] and functional richness [47] for all sites and species pools in a single trait space (figure 1) using the dbFD function from the 'FD' R package [48]. The traits were weighted such that all six traits contributed equally to both calculations. Functional dispersion measures the distribution of species in trait space or how similar a community is in terms of its functional traits. Functional dispersion can either be unweighted, with all species accounted for equally, or weighted by species abundances, with distance from the community centroid to abundant species in trait space contributing more to the metric than distances from the community centroid to rare species. Functional richness measures the volume of the convex hull encompassing all species in a community in trait space. The functional richness metric is incapable of taking species abundances into account and neither metric can infer mechanistic links between individual traits and environmental drivers.

We quantified species pool functional richness, habitat heterogeneity, primary productivity and two measures of anthropogenic disturbance to use as predictors of functional richness and occupancy-weighted functional dispersion. The species pool included all forest-dwelling, terrestrial species over 1 kg that may inhabit the park based on their geographical ranges, which we extracted for each protected area using global mammal ranges from the IUCN Redlist [49]. Species pool functional richness was

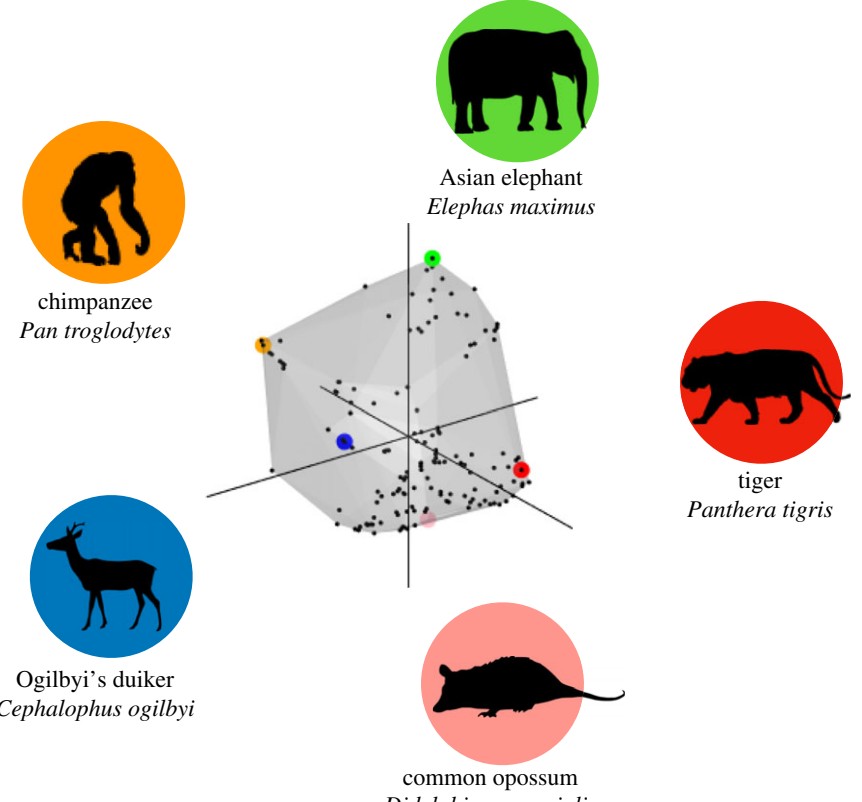

**Figure 1.** Total trait space occupied by species found in realized communities of all TEAM sites. Grey shading indicates the three-dimensional convex hull of this trait space, which represents the first three axes used in the functional diversity calculations. Black points represent species. Example species represented by coloured dots with silhouettes correspond with coloured circles in trait space. Opossum image created by Sarah Werning, licence available at https://creativecommons.org/licenses/by/3.0/. (Online version in colour.)

calculated using the same function as realized community functional richness. We calculated habitat heterogeneity as the Shannon diversity of landcover types within a 2 km buffer of the minimum convex polygon surrounding the camera traps at each site. We obtained land cover types from Copernicus satellite data from 2015 and followed classifications according to the Copernicus Global Land Service Dynamic Landcover Map at a 100 m resolution (CGLS-LC100, [50]). We used published primary productivity values for TEAM sites from Rovero *et al.* [38], specifically the normalized difference vegetation index (NDVI). We also used Rovero *et al.* [38] values for human density within the zone of interaction, which incorporates land cover, elevation, waterways and anthropogenic developments to delineate the spatial extent expected to impact natural processes inside the protected area [51]. Finally, we included the impact of local species extinction on functional richness as an additional measure of anthropogenic disturbance and as an offset to control for losses in functional richness due to extirpations that have been documented at four TEAM sites over recent decades. Specifically, Korup, Cameroon, lost the leopard, golden cat and giant pangolin; Barro Colorado Island and Soberania National Park, Panama, lost the white-lipped peccary, giant anteater and likely the jaguar; Bwindi Impenetrable Forest, Uganda, lost the buffalo, leopard and giant forest hog; Nam Kading, Laos, lost both the tiger and leopard [38]. We note that local extinctions may not necessarily impact functional diversity because of the functional redundancy that occurs in tropical forest mammal communities [52]. We quantified local extinction for each TEAM site as the difference in functional richness between the functional richness of the current realized community and the functional richness of current realized community as well as the recently extirpated species. For ranges of protected area-level characteristics among sites, see electronic supplementary material, table S4.

## (c) Analysis

We modelled occupancy-weighted functional dispersion and community functional richness as a function of environmental and anthropogenic predictor variables using Bayesian generalized linear regression. None of the predictor variables were highly correlated ($r < 0.6$) and all continuous variables were scaled and centred to produce standardized $\beta$ coefficient estimates with a mean of zero and standard deviation of 1. Given the left-skew in the distribution of both functional diversity metrics, we specified a Weibull distribution for both of our models. The global model consisted of all predictor variables described above as well as a categorical fixed effect for biogeographic region (Neotropics, Africa, Asia, Madagascar) to account for region-specific patterns in functional diversity. We used the brm function from the brms [53] package in R to fit the models. We visually assessed model trace plots (electronic supplementary material, figure S1), used Rhat criteria (less than 1.05) to assess convergence and performed graphical posterior predictive checks on model distributions and error (electronic supplementary material, figure S2). We interpreted a significant contribution of a predictor variable to functional diversity if the 95% credible interval of a parameter's posterior distribution did not include zero.

To assess the importance of occupancy in functional dispersion, we also ran regression models for two additional unweighted functional dispersion calculations. Unlike occupancy-weighted functional dispersion, the additional calculations treated all species equally and did not incorporate variation in species abundances. As a result, they likely omit important ecological information about the relative abundances of functional traits. We calculated unweighted functional dispersion using (i) presence data of the observed community from camera trap data, and separately using (ii) presence data extracted from IUCN range maps. See electronic supplementary material, appendix S10 for full materials and methods.

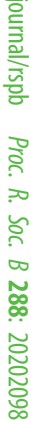

**(a)** global functional dispersion **(b)**

**(c)** global functional richness **(d)**

**Figure 2.** Results of global tropical mammal functional richness and functional dispersion regression models. Credible interval plots show the effect size of predictors of occupancy-weighted functional dispersion (*a*) and functional richness (*c*). The distribution indicates the full posterior of the parameter. Points indicate medians, thin black lines indicate 95% credible intervals and thick black lines indicate 50% credible intervals. Dark grey β effect distributions indicate a non-zero 95% credible interval, while light grey β effect distributions indicate a 95% credible interval that includes zero. All predictor variables were standardized to make their estimates comparable. The intercept effects, which are outputs of the Weibull models, represent the log-transformed functional diversity measurement of the baseline biogeographic region (i.e. the Neotropics) when all other covariates are held at zero. Maps show occupancy-weighted functional dispersion (*b*) and functional richness (*d*) for each of the TEAM study sites. Coloured shading indicates the three-dimensional convex hull of the trait space occupied by the realized mammal community, using the first three axes from the functional diversity calculation. Black dots represent the location in trait space of individual species, and the dot size represents occurrence probability (in *b*). Shape colour indicates values of occupancy-weighted functional dispersion (*b*) value or the functional richness (*d*). Large grey circles (in *d*) indicate species that were extirpated in recent decades and were not part of the functional richness calculation. The species positioned exterior to the convex hull demonstrate how their extirpation caused a decrease in community trait space. All shapes are directly comparable because they were plotted in the same trait space and have the same conformation in trait space. (Online version in colour.)

## 3. Results

The regression model predicting occupancy-weighted functional dispersion produced a significant effect for NDVI ($\beta = 0.07$, Err = 0.03) with a 95% credible interval that did not overlap zero (figure 2*a*), which indicates that the occupancies of species with unique functional traits increased as productivity increased. Communities at lower productivity sites comprised species with high occupancy clustered together in trait space (e.g. figure 2*b* Volcán Barva), indicating that a large proportion of individuals within the community had similar functional traits to each other. Communities at high productivity sites had species with high occupancies

distributed throughout trait space (e.g. figure 2*b* Barro Colorado), meaning that many individuals within the community had functional traits that were relatively unique.

In contrast with the significant, positive association of occupancy-weighted functional dispersion of the realized mammal community with NDVI, neither of the unweighted functional dispersion measures (i.e. using presence-only data based on observed species or based on IUCN range maps) were significantly associated with NDVI (figure 3).

All four TEAM sites that experienced recent local extinctions showed declines in functional richness. Specifically, Korup lost 33.4% ($3.9 \times 10^{-3}$ units) of its functional richness, Barro Colorado lost 7% ($6.9 \times 10^{-4}$ units), Bwindi lost 4.6% ($6.8 \times 10^{-4}$

Proc. R. Soc. B 288: 20202098

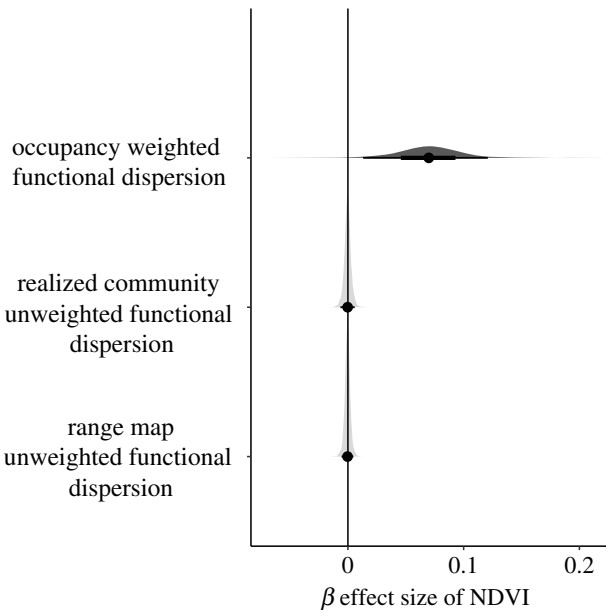

**Figure 3.** Credible intervals showing the effect size of the NDVI parameter on three different measures of functional dispersion. The distribution represents the full posterior of the NDVI parameter from three different generalized linear regression models predicting (i) occupancy-weighted functional dispersion, which was calculated using occupancy estimates from the camera traps, (ii) realized community-unweighted functional dispersion, which used presence-only data of the realized community observed by camera traps in each site, and (iii) species pool unweighted functional dispersion, which was calculated for the mammal community defined by presence-only data of species assumed to occur at the site based on range map overlap with the protected area. Dark grey distributions indicate a non-zero 95% credible interval, while light grey distributions indicate the 95% credible interval includes zero.

units) and Nam Kading lost 7.2% ($1.1 \times 10^{-3}$ units). The regression model predicting community functional richness contained a significant negative effect for local extinction ($\beta = -0.18$, Err = 0.06, figure 2c), which indicates that human-induced extirpations have had a significant impact on tropical mammal functional richness. There was also a significant negative effect for Madagascar ($\beta = -2.60$, Err = 0.47), indicating that functional richness was significantly lower at the Malagasy site than sites in other regions.

To assess whether weighting traits equally in the dbFD function affected the directionality and significance of the predictors of functional diversity, we compared model results with and without equal weighting of traits. Functional richness was unaffected by trait weighting, but functional dispersion without trait weighting had additional significant predictors. Specifically, local extinctions and the Madagascar bioregion emerged as significant negative predictors of functional dispersion when weights to treat traits equally were excluded (electronic supplementary material, figure S3). Notably, these additional predictors were also significant predictors of functional richness. We retained the weights, which considered all traits equally, to adhere to the precedent established in previous studies of mammal functional diversity [28,35,37].

## 4. Discussion

We tested four non-mutually exclusive hypotheses to assess the importance of historical, environmental and anthropogenic factors in determining the observed functional diversity of mammal communities in tropical forest protected areas around the world. We found that occupancy-weighted functional dispersion was positively associated with productivity, while functional richness was negatively associated with local extinctions and was lower in Madagascar than other tropical regions. The lack of significant effects for Africa, Asia and the Neotropics suggests that the effects of productivity and local extinction were similarly influential across very different systems and, therefore, that consistent mechanisms may be operating across diverse tropical regions. Our results failed to support the role of habitat heterogeneity in driving functional richness or occupancy-weighted functional dispersion, which may reflect that landcover classifications were too coarse to capture habitat heterogeneity at scales relevant to tropical forest mammals or that the spatial arrangement of habitat types rather than their composition could affect functional diversity. Species pools were also not a statistically significant driver of either functional diversity metric, but did have a positive effect on functional richness. The sample size in this study may have been too small to detect a significant effect of species pools. Nevertheless, the non-significant positive effect suggests that larger species pool functional richness may facilitate larger realized community functional richness if it increases the likelihood that a pool contains species with functional traits that can occupy all available niches in a community [1]. Thus, we found that global variation in tropical mammal functional diversity can be attributed to energetic and anthropogenic influences and varies among biogeographic regions likely due to unique historical and evolutionary influences.

## (a) Functional dispersion

Our model of occupancy-weighted functional dispersion identified the importance of productivity in tropical mammal functional dispersion. The strong positive relationship lends support to the niche diversity mechanism of diversity–productivity relationships, in which higher productivity leads to greater success of species with diverse niches, represented here by unique functional traits. According to this proposed mechanism, high productivity increases the availability of all resources, but disproportionally favours species based on the uniqueness of their functional traits and the resources they exploit [25]. Species with unique functional traits can persist at higher relative abundances because they face less interspecific competition than species with more typical traits [54]. Our results suggest that as productivity increases, species with unique traits are relatively more abundant in tropical forest mammal communities, resulting in an increase in occupancy-weighted functional dispersion.

Notably, the positive relationship between occupancy-weighted functional dispersion and productivity differs from global-scale relationships between unweighted mammal functional dispersion and productivity [28,36]. Multiple studies have reported low mammal functional dispersion in tropical rainforests, which is typically attributed to niche packing and high functional redundancy [35,36]. Taken together, the high productivity of tropical rainforests may support species that are more functionally redundant than in other biomes while also supporting higher abundances of species with unique functional traits.

Measures of functional diversity that incorporate occupancy or abundance from observational data provide important information beyond what is possible from range

map-based studies because weighting species by their relative abundances details how functions are distributed among individuals in a community from population-level data. When we tested whether occupancy affected the relationship between functional dispersion and NDVI, we found that functional dispersion was only significantly associated with NDVI when species occupancies were incorporated. Standardized *in situ* camera trap data collected at a global scale have, therefore, provided insight into community assembly processes that could not be identified from species lists or range map data alone.

## (b) Functional richness

Our results indicate that recent, human-induced local extinctions in part determined mammal functional richness in the tropical forest protected areas we surveyed. The relatively high degree of functional redundancy that occurs in tropical mammal communities has the potential to prevent local extinctions from decreasing functional diversity [52]. Therefore, the significant negative effect of known recent extirpations on functional richness is an important finding because it indicates that local species losses have significantly reduced the functional richness of tropical forest mammal communities. Declines in functional diversity may in turn impact ecosystem functioning. Notably, sites that experienced local extinctions all lost their top predator. Previous research has shown that top predators are often extirpated first from systems under anthropogenic pressure, and that their loss often has disproportionate effects on the ecosystem due to cascading impacts on lower trophic levels and ecosystem services [55,56]. Korup, which lost its top predator as well as its only other obligate carnivore (African golden cat) and a unique specialist species (giant pangolin), had the greatest decline in functional richness. Our study reinforces that the loss of top predators and specialists will likely result in disproportionate losses of functional diversity as anthropogenic influences continue to alter tropical mammal communities [32,33]. Lastly, the negative effect of Madagascar on functional richness is likely a result of the unique biological and geological history of Madagascar [57]. The fact that the single Malagasy TEAM site, Ranomafana, had substantially lower species richness [58] and functional diversity than TEAM sites in other biogeographic regions may also be partly attributable to the small size and insular nature of Madagascar.

## 5. Conclusion

This study suggests that higher productivity in tropical moist forests may allow greater success of mammal species with diverse niches and support more functionally diverse communities as a result. At the same time, anthropogenically induced extirpations may be causing significant declines in functional diversity in these communities, despite their high functional redundancy. Based on these findings, we strongly encourage the continuation and expansion of standardized monitoring studies to allow a greater understanding of community assembly and disassembly in the Anthropocene.

Data accessibility. All data used in this project are published in the Dryad Digital Repository at https://doi.org/10.5061/dryad.f1vhhmgv0 [59]. All code used in the analysis and figuremaking are published on github at https://github.com/DanielGorczynski/Global-Functional-Diversity. Methods for occupancy estimates are available in Beaudrot *et al*. [40]. Methods for calculating additional site-level predictor variables are available in Rovero *et al*. [38]. Raw camera trap data from the TEAM Network are available at Wildlife Insights.

Authors' contributions. D.G. provided conceptualization, data collection (trait data), data processing and analysis, initial drafting and revision of this manuscript. C.H. and J.T.L. performed data acquisition and analysis on the species pools and habitat heterogeneity variables as well as provided feedback and revision on the draft. J.A., S.E., S.J., F.R. and F.S. provided data collection, management and processing and provided significant feedback on manuscript draft. M.H.A., J.H.A., P.A.J., C.K., M.G.M.L. and J.S. provided data collection, management and processing. L.B. provided project supervision, conceptualization, writing and significant editing for the project. All authors gave final approval for publication and agree to be held accountable for the work performed.

Competing interests. We declare we have no competing interests.

Funding. This study was partially funded by these institutions: the Gordon and Betty Moore Foundation, HP, Northrop Grumman Foundation and other donors.

Acknowledgements. All TEAM data were provided by the Tropical Ecology Assessment and Monitoring (TEAM) Network, a collaboration between Conservation International, the Smithsonian Institute and the Wildlife Conservation Society. We thank all TEAM staff and affiliates. We also thank Dr Amy Dunham, Dr Volker Rudolf, Dr Daniel Kowal, Dr Tom E.X. Miller and Carsten Grupstra for feedback and discussion.

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
