## [Peer Review File · Proceedings of the Royal Society B: Biological Sciences]

Review History

RSPB-2020-2098.R0 (Original submission)

Review form: Reviewer 1 (Brunno Oliveira)

Recommendation

Accept with minor revision (please list in comments)

Scientific importance: Is the manuscript an original and important contribution to its field?

Good

General interest: Is the paper of sufficient general interest?

Good

Quality of the paper: Is the overall quality of the paper suitable?

Excellent

Is the length of the paper justified?

Yes

Should the paper be seen by a specialist statistical reviewer?

No

Do you have any concerns about statistical analyses in this paper? If so, please specify them explicitly in your report.

No

It is a condition of publication that authors make their supporting data, code and materials available - either as supplementary material or hosted in an external repository. Please rate, if applicable, the supporting data on the following criteria.

Is it accessible?

Yes

Is it clear?

Yes

Is it adequate?

Yes

Do you have any ethical concerns with this paper?

No

Comments to the Author

My comments are attached (See Appendix A)

Review form: Reviewer 2

Recommendation

Major revision is needed (please make suggestions in comments)

Scientific importance: Is the manuscript an original and important contribution to its field?

Good

General interest: Is the paper of sufficient general interest?

Good

Quality of the paper: Is the overall quality of the paper suitable?

Good

Is the length of the paper justified?

Yes

Should the paper be seen by a specialist statistical reviewer?

Yes

Do you have any concerns about statistical analyses in this paper? If so, please specify them explicitly in your report.

Yes

It is a condition of publication that authors make their supporting data, code and materials available - either as supplementary material or hosted in an external repository. Please rate, if applicable, the supporting data on the following criteria.

Is it accessible?

Yes

Is it clear?

Yes

Is it adequate?

Yes

Do you have any ethical concerns with this paper?

No

Comments to the Author

General comments: Gorczynski et al. tested 4 hypotheses – species pool hypothesis, spatial heterogeneity, primary production, and disturbance – for what drives functional diversity and dispersion. The present a clear question and approach coupled with a localized approach to estimating community composition via camera trap surveys. I think this is a nice, clean paper that tests multiple competing hypotheses and tells a compelling story. However, I have some general analytical concerns that could take significant work to overcome, and I have listed a number of specific comments to help improve clarity as well.

1. My biggest concern is the linear regressions. I ran the code and saw what appear to be a number of outliers as well as a potential lack of heteroscedasticity in residual variance. With only 15 data points I don't think this is particularly surprising but needs to be addressed more thoroughly. Classic approaches like transformations could help with residuals, but outliers with leverage will be harder to account for. I might recommend a Bayesian linear regression that can better assess model fits through leave-one-out cross validation that directly account for outliers. These models are easy to implement in brms and rstanarm (among others) with identical syntax to lm/lme4.

2. While limited in sample size, I think (and the authors note) that there are differences between regions in both response and predictors. This lends itself to a LMM approach with region/continent as a random effect to account for the underlying variation and hierarchical nature of clustered sampling/sites. While difficult to address with a standard LM/LMM approach, this would certainly be doable in a Bayesian framework. In general it looks like the data is good and the story is interesting, but the models are limiting.

Specific comments:

L37-43 – The abstract doesn't present any true results and I think it could use something here. What is the role/directionality of human presence, productivity, distance to settlement, etc?

L80-85 – Move respectively to end of sentence. Also consider breaking this up. The mechanisms are important but get lost in these complex sentences.

L89-97 – In a classical sense the role of disturbance in mediating diversity is similar to heterogeneity in that it creates opportunities for niche partitioning and mediates coexistence thereby increasing diversity. But human disturbances are extremely non-random and have known directionality. I think that needs some more unpacking here, as human disturbance likely does not have the same predicted effect as other forms of disturbance (e.g., fire).

L130 – Ground-dwelling per se or simply terrestrial (non-arboreal) mammals?

L134 – Clarify that this is citation 40 and not data from 40 sites – is that right?

L139 – Table 1 never cited, should go in this paragraph. A few additional details would also help: how was body mass calculated? Were activity and habitat in Table determined by cam data or lit review? Same with litter size. If these were means drawn from the citations that should be explicit. Also I think "habitat" is more like "arboreality."

L157-158 – Is this distance from centroid of MCP or mean distance from camera to settlement?

L161-163 – This local extinction is really more like the functional beta diversity which makes this circular no? As a predictor it measures the effect of changing functional richness on functional richness?

L166-173 – Are the sites independent? It seems to me that you have 3 distinct regions – South America, Africa, and Southeast Asia – that could also have their own structure, especially with respect to predictors like species pools, NPP, and disturbance. I think a mixed effects model would be a better application here as it would not only account for this underlying variation but would also tell you if specific regions diverge from the overall pattern presented. This would provide important information for regional management and conservation that is currently not represented.

L174-179 – This needs more explanation. The intro emphasizes the importance of local occupancy for calculating site level FD/FR, but its buried in the methods and this is the chance to show that utility. I'm also not entirely sure what was done here. Occupancy was used to weight FD/FR (how? Basic description would better than a citation in my opinion), and then calculated unweighted using camera data and range maps (classical approach)? A few extra lines outlining this would provide more clarity and help future researchers recreate your approach, which I would guess is a primary objective here.... I now see the full materials and methods in S11, but it's never cited. A condensed methods is fine, but the key points need to be clearer and the full methods need to be cited so anyone can go check them.

L182-194 – I'm generally not a fan of tables, but this whole paragraph is summary stats that could be summarized more efficiently in a table and would save a lot of text for more important info (e.g., additional methods context).

L196 – I would use beta/ β instead of est

L195-208 – I ran the code and checked the models because I was skeptical about only 15 data points detecting significance of multiple predictors. Between the Shapiro-test and the QQ plot its clear, as noted, that the residuals are normally distributed, but a fitted vs residual plot shows a pretty clear lack of heteroscedasticity (larger fitted values have much more variance than smaller). It also looks like there are a number of outliers for most of the predictors, and I would be worried about the leverage these have given the small sample size. I think some transformations are likely in order, or a Bayesian linear regression that can handle complex models with limited data (these are now very easy to implement in packages like brms and rstanarm that have the same syntax as lm/lme4). I'd also note that neither Fdis nor Frich look like they are normal distributions, so a transformation or a generalized lm with an alternative distribution might help with this.

L251-254 – This is interesting because landcover.div vs. Frich in the data looks like an exponential pattern. It's also possible that heterogeneity in the form of landscape shape, not just composition, could drive functional richness.

L268-269 – This is very interesting, and again I think I mixed-effects structure would help pull out this effect and quantify it. I would also be interested in the interaction between species pools and human disturbance, though you might be data limited in terms of sites.

L293-297 – Which traits are driving this pattern? I think this is true, but as with most functional traits at this coarse resolution (binary indicators of list history traits) its hard to tie an actual function or niche to them. It would be nice to show a mechanism. You talk about this limitation below, but being able to tie some of these potential consequences to species.

Decision letter (RSPB-2020-2098.R0)

09-Nov-2020

Dear Mr Gorczynski:

Your manuscript has now been peer reviewed and the reviews have been assessed by an Associate Editor. The reviewers' comments (not including confidential comments to the Editor) and the comments from the Associate Editor are included at the end of this email for your reference. As you will see, the reviewers and the Editors have raised some concerns with your manuscript and we would like to invite you to revise your manuscript to address them.

Research ethics:

Use of animals and field studies:

It is a condition of publication that you make available the data and research materials supporting the results in the article. Please see our Data Sharing Policies (<https://royalsociety.org/journals/authors/author-guidelines/#data>). Datasets should be deposited in an appropriate publicly available repository and details of the associated accession number, link or DOI to the datasets must be included in the Data Accessibility section of the

article (<https://royalsociety.org/journals/ethics-policies/data-sharing-mining/>). Reference(s) to datasets should also be included in the reference list of the article with DOIs (where available).

Please submit a copy of your revised paper within three weeks. If we do not hear from you within this time your manuscript will be rejected. If you are unable to meet this deadline please let us know as soon as possible, as we may be able to grant a short extension.

Best wishes,
Dr Locke Rowe
mailto: proceedingsb@royalsociety.org

Editor

Comments to Author(s):

This manuscript has now been reviewed by two experts in the field. Their extensive comments are in general quite positive. Both referees see much value in the work, as do I. However, both have concerns about the appropriateness of the analyses and therefore strength of the results. In light of these, I am suggesting the ms be revised if the authors believe that after exploring alternative analyses the main results stand.

The two main issues in the analyses are around the use of regression. First, one referee believes that some analyses are highly leveraged, and suggests alternative approaches to explore the impact of this on the results. Second, there was a suggestion to take a more mixed model approach to address region/continental effects. Several other suggestions are made by both referees to improve the analyses.

The referees have also made numerous other suggestions aimed at improving clarity. One that I would like to emphasize is that the results section is too long, resulting in key messages being lost. The referees have made suggestions to address this. I would like to add that I think the discussion is also unnecessarily long.

Reviewer(s)' Comments to Author:

Referee: 1

Comments to the Author(s)

My comments are attached

Referee: 2

Comments to the Author(s)

General comments: Gorczynski et al. tested 4 hypotheses – species pool hypothesis, spatial heterogeneity, primary production, and disturbance – for what drives functional diversity and dispersion. The present a clear question and approach coupled with a localized approach to estimating community composition via camera trap surveys. I think this is a nice, clean paper that tests multiple competing hypotheses and tells a compelling story. However, I have some general analytical concerns that could take significant work to overcome, and I have listed a number of specific comments to help improve clarity as well.

1. My biggest concern is the linear regressions. I ran the code and saw what appear to be a number of outliers as well as a potential lack of heteroscedasticity in residual variance. With only 15 data points I don't think this is particularly surprising but needs to be addressed more thoroughly. Classic approaches like transformations could help with residuals, but outliers with leverage will be harder to account for. I might recommend a Bayesian linear regression that can better assess model fits through leave-one-out cross validation that directly account for outliers. These models are easy to implement in brms and rstanarm (among others) with identical syntax to lm/lme4.

2. While limited in sample size, I think (and the authors note) that there are differences between regions in both response and predictors. This lends itself to a LMM approach with region/continent as a random effect to account for the underlying variation and hierarchical nature of clustered sampling/sites. While difficult to address with a standard LM/LMM approach, this would certainly be doable in a Bayesian framework. In general it looks like the data is good and the story is interesting, but the models are limiting.

Specific comments:

L37-43 – The abstract doesn't present any true results and I think it could use something here. What is the role/directionality of human presence, productivity, distance to settlement, etc?

L80-85 – Move respectively to end of sentence. Also consider breaking this up. The mechanisms are important but get lost in these complex sentences.

L89-97 – In a classical sense the role of disturbance in mediating diversity is similar to heterogeneity in that it creates opportunities for niche partitioning and mediates coexistence thereby increasing diversity. But human disturbances are extremely non-random and have known directionality. I think that needs some more unpacking here, as human disturbance likely does not have the same predicted effect as other forms of disturbance (e.g., fire).

L130 – Ground-dwelling per se or simply terrestrial (non-arboreal) mammals?

L134 – Clarify that this is citation 40 and not data from 40 sites – is that right?

L139 – Table 1 never cited, should go in this paragraph. A few additional details would also help: how was body mass calculated? Were activity and habitat in Table determined by cam data or lit

review? Same with litter size. If these were means drawn from the citations that should be explicit. Also I think “habitat” is more like “arboreality.”

L157-158 – Is this distance from centroid of MCP or mean distance from camera to settlement?

L161-163 – This local extinction is really more like the functional beta diversity which makes this circular no? As a predictor it measures the effect of changing functional richness on functional richness?

L166-173 – Are the sites independent? It seems to me that you have 3 distinct regions – South America, Africa, and Southeast Asia – that could also have their own structure, especially with respect to predictors like species pools, NPP, and disturbance. I think a mixed effects model would be a better application here as it would not only account for this underlying variation but would also tell you if specific regions diverge from the overall pattern presented. This would provide important information for regional management and conservation that is currently not represented.

L174-179 – This needs more explanation. The intro emphasizes the importance of local occupancy for calculating site level FD/FR, but its buried in the methods and this is the chance to show that utility. I’m also not entirely sure what was done here. Occupancy was used to weight FD/FR (how? Basic description would better than a citation in my opinion), and then calculated unweighted using camera data and range maps (classical approach)? A few extra lines outlining this would provide more clarity and help future researchers recreate your approach, which I would guess is a primary objective here... I now see the full materials and methods in S11, but it’s never cited. A condensed methods is fine, but the key points need to be clearer and the full methods need to be cited so anyone can go check them.

L182-194 – I’m generally not a fan of tables, but this whole paragraph is summary stats that could be summarized more efficiently in a table and would save a lot of text for more important info (e.g., additional methods context).

L196 – I would use β instead of *est*

L195-208 – I ran the code and checked the models because I was skeptical about only 15 data points detecting significance of multiple predictors. Between the Shapiro-test and the QQ plot its clear, as noted, that the residuals are normally distributed, but a fitted vs residual plot shows a pretty clear lack of heteroscedasticity (larger fitted values have much more variance than smaller). It also looks like there are a number of outliers for most of the predictors, and I would be worried about the leverage these have given the small sample size. I think some transformations are likely in order, or a Bayesian linear regression that can handle complex models with limited data (these are now very easy to implement in packages like brms and rstanarm that have the same syntax as lm/lme4). I’d also note that neither *Fdis* nor *Frich* look like they are normal distributions, so a transformation or a generalized lm with an alternative distribution might help with this.

L251-254 – This is interesting because *landcover.div* vs. *Frich* in the data looks like an exponential pattern. It’s also possible that heterogeneity in the form of landscape shape, not just composition, could drive functional richness.

L268-269 – This is very interesting, and again I think I mixed-effects structure would help pull out this effect and quantify it. I would also be interested in the interaction between species pools and human disturbance, though you might be data limited in terms of sites.

L293-297 – Which traits are driving this pattern? I think this is true, but as with most functional traits at this coarse resolution (binary indicators of list history traits) its hard to tie an actual

function or niche to them. It would be nice to show a mechanism. You talk about this limitation below, but being able to tie some of these potential consequences to species.

Author's Response to Decision Letter for (RSPB-2020-2098.R0)

See Appendix B.

RSPB-2020-2098.R1 (Revision)

Review form: Reviewer 2

Recommendation

Accept with minor revision (please list in comments)

Scientific importance: Is the manuscript an original and important contribution to its field?

Good

General interest: Is the paper of sufficient general interest?

Excellent

Quality of the paper: Is the overall quality of the paper suitable?

Good

Is the length of the paper justified?

Yes

Should the paper be seen by a specialist statistical reviewer?

No

Do you have any concerns about statistical analyses in this paper? If so, please specify them explicitly in your report.

No

It is a condition of publication that authors make their supporting data, code and materials available - either as supplementary material or hosted in an external repository. Please rate, if applicable, the supporting data on the following criteria.

Is it accessible?

Yes

Is it clear?

Yes

Is it adequate?

Yes

Do you have any ethical concerns with this paper?

No

Comments to the Author

RSPB-2020-2098.R1 - "Tropical mammal functional diversity increases with productivity but decreases with anthropogenic disturbance"

Gorcynzski et al. have done a nice job with reviewer suggestions and I feel that the ms is much improved. In particular, I believe the transition to a Bayesian regression approach is much more appropriate, and I fully agree that a single, global model testing the relative effects of each competing hypothesis is the way to go here. I also like the added emphasis on occupancy-weighted measurements compared to range-based approaches, as it shows the novelty and added utility of the approach presented herein. My remaining comments are largely editorial and aimed at clarifying the ms. Nicely done!

Specific Comments (line numbers from tracked changes ms in Response):

L101-102 - "increasing the energetic capacity of an ecosystem and allowing greater energy allocation to individual organisms" - I think this would read better as "presumably due to increased energetic capacity..." As mentioned, there are multiple mechanisms in play and the diversity-productivity debate is not especially settled.

L129 - affect

L131-133 - This sentence reads oddly. I think it should be "the relative strength of anthropogenic disturbance on tropical mammal community assembly" or "the relative importance of anthropogenic disturbance in tropical mammal community assembly." Also, I think you need to clarify what the strength is relative to. Other regions? Other drivers? Given that you are comparing 4 hypotheses, I think it's the latter, but the subject of the sentence (gradient across sites) suggests the former.

L139 - Temperate communities

L204-205 - I think some discussion of why these traits matter is needed. Are they response or effect traits? The introduction mentions the importance of mammals and their trait variation for ecosystems, this would be the spot to unpack that (or in the table). Too many macroecological studies lose sight of the actual functions associated with functional traits. I don't think that's the case here as most traits do map to, or scale with, specific functions/services...but some justification of why these traits are representative of functional diversity and why we should care about changes to them would be good.

L276-288 - I think this is a much-improved framework to test competing hypotheses with limited data. In addition to manual trace examinations, you should also present (supplementary) posterior predictive checks of your errors and distributions to show that your models accurately reflect the data - see here (<https://mc-stan.org/bayesplot/reference/PPC-overview.html#ppc-plotting-functions>). As is, it is still not clear whether the model fits are good, especially since no model selection was used to assess fits (though I agree with your decision to use a single, global model).

L324 - One of the best things about a Bayesian approach is that it frees you from the confines of statistical significance in the classical sense. I think it's fine to use credible intervals not overlapping zero as a measure of significance, but the Bayesian framework also allows you to assess the relative strengths of your multiple competing hypotheses even if they are not significant per se...e.g., extinction and site (Madagascar) had "significant" effects on richness, but even when accounting for those the effect of species pools was still positive and substantial, just not as strong. That seems notable, and given the relatively small sample size used here it's not unreasonable to think you simply lacked the power or variation to detect a stronger effect.

L509 - Change to "Our results indicate that recent, human-induced local extinctions determined mammal functional richness in the tropical forest protected areas we surveyed." ... improve concision

L519-521 - The loss of predators is indeed notable, but not surprising (see/cite Estes et al. 2011, Ripple et al. 2014, Science). Carnivores have disproportionate effects on ecosystem processes, and the loss of these species and their associated traits likely has cascading impacts on the rest of the system. Your results emphasize that point. An interesting way to show this would be highlighting extirpated species in Fig 1 to illustrate how these species impact trait space.

L521-524 - This is the only real mention of geographic region. As you note, Madagascar has a very unique phylogeographic history which likely drives the pattern. It is also the only island, has only 1 site, and is by far the smallest of the regions. All of these also likely impact functional

diversity (and probably should be noted). More notably though, is that the other regions do not have strong effects, suggesting that the important drivers of functional diversity – primary production and anthropogenic disturbance – and are consistent across very different systems. From a basic ecology perspective that is an interesting, and suggests a consistent mechanism which you highlight (niche packing).

L526 – I think this limitations section is distracting and detracts from the study. As currently presented, it is simply a list of potential issues that could undermine results. I think it needs one of two approaches: 1) address each point and why the study still “holds.” i.e., defend the ms against these critiques. This may be difficult and will take space. 2) Delete the paragraph entirely and distribute throughout ms. Most of these are methodological and could be addressed in a sentence following the introduction of the approaches. These are mostly known limitations, and a simple justification of the approach after introducing it would be sufficient IMO.

L634 – This conclusion only highlights one of two primary results, that occupancy-weighting detected response to NDVI. The other – human impacts – is noted in the title and should be highlighted here too. Moreover, these conclusions are more of a summary and could likely be cut entirely. My suggestion is to move the “limitations” throughout the ms (as noted above, #2) and use this conclusion to briefly highlight and then make suggestions for future work.

Fig 2 – Two points: 1) Should plot the intercept – in this case South America – so that all sites are represented. Right now it is just the betas. 2) Do the 3D FD plots include the extirpated species? As noted above, I think illustrating which species are lost and how that changes trait space would be interesting and help the reader understand how extinctions drive functional traits.

Decision letter (RSPB-2020-2098.R1)

14-Jan-2021

Dear Mr Gorczynski

I am pleased to inform you that your manuscript RSPB-2020-2098.R1 entitled "Tropical mammal functional diversity increases with productivity but decreases with anthropogenic disturbance" has been accepted for publication in Proceedings B.

The referee(s) have recommended publication, but also suggest some minor revisions to your manuscript. Therefore, I invite you to respond to the referee(s)' comments and revise your manuscript. Because the schedule for publication is very tight, it is a condition of publication that you submit the revised version of your manuscript within 7 days. If you do not think you will be able to meet this date please let us know.

Sincerely,
 Dr Locke Rowe
 Editor, Proceedings B
 mailto:proceedingsb@royalsociety.org

Reviewer(s)' Comments to Author:

Referee: 2

Comments to the Author(s)

RSPB-2020-2098.R1 - "Tropical mammal functional diversity increases with productivity but decreases with anthropogenic disturbance"

Gorcynzski et al. have done a nice job with reviewer suggestions and I feel that the ms is much improved. In particular, I believe the transition to a Bayesian regression approach is much more appropriate, and I fully agree that a single, global model testing the relative effects of each competing hypothesis is the way to go here. I also like the added emphasis on occupancy-weighted measurements compared to range-based approaches, as it shows the novelty and added utility of the approach presented herein. My remaining comments are largely editorial and aimed at clarifying the ms. Nicely done!

Specific Comments (line numbers from tracked changes ms in Response):

L101-102 - "increasing the energetic capacity of an ecosystem and allowing greater energy allocation to individual organisms" - I think this would read better as "presumably due to increased energetic capacity..." As mentioned, there are multiple mechanisms in play and the diversity-productivity debate is not especially settled.

L129 - affect

L131-133 - This sentence reads oddly. I think it should be "the relative strength of anthropogenic disturbance on tropical mammal community assembly" or "the relative importance of anthropogenic disturbance in tropical mammal community assembly." Also, I think you need to clarify what the strength is relative to. Other regions? Other drivers? Given that you are comparing 4 hypotheses, I think it's the latter, but the subject of the sentence (gradient across sites) suggests the former.

L139 - Temperate communities

L204-205 - I think some discussion of why these traits matter is needed. Are they response or effect traits? The introduction mentions the importance of mammals and their trait variation for ecosystems, this would be the spot to unpack that (or in the table). Too many macroecological studies lose sight of the actual functions associated with functional traits. I don't think that's the case here as most traits do map to, or scale with, specific functions/services...but some justification of why these traits are representative of functional diversity and why we should care about changes to them would be good.

L276-288 - I think this is a much-improved framework to test competing hypotheses with limited data. In addition to manual trace examinations, you should also present (supplementary) posterior predictive checks of your errors and distributions to show that your models accurately reflect the data - see here (<https://mc-stan.org/bayesplot/reference/PPC-overview.html#ppc-plotting-functions>). As is, it is still not clear whether the model fits are good, especially since no model selection was used to assess fits (though I agree with your decision to use a single, global model).

L324 - One of the best things about a Bayesian approach is that it frees you from the confines of statistical significance in the classical sense. I think it's fine to use credible intervals not overlapping zero as a measure of significance, but the Bayesian framework also allows you to assess the relative strengths of your multiple competing hypotheses even if they are not significant per se...e.g., extinction and site (Madagascar) had "significant" effects on richness, but even when accounting for those the effect of species pools was still positive and substantial, just

not as strong. That seems notable, and given the relatively small sample size used here it's not unreasonable to think you simply lacked the power or variation to detect a stronger effect.

L509 - Change to "Our results indicate that recent, human-induced local extinctions determined mammal functional richness in the tropical forest protected areas we surveyed." ... improve concision

L519-521 - The loss of predators is indeed notable, but not surprising (see/cite Estes et al. 2011, Ripple et al. 2014, Science). Carnivores have disproportionate effects on ecosystem processes, and the loss of these species and their associated traits likely has cascading impacts on the rest of the system. Your results emphasize that point. An interesting way to show this would be highlighting extirpated species in Fig 1 to illustrate how these species impact trait space.

L521-524 - This is the only real mention of geographic region. As you note, Madagascar has a very unique phylogeographic history which likely drives the pattern. It is also the only island, has only 1 site, and is by far the smallest of the regions. All of these also likely impact functional diversity (and probably should be noted). More notably though, is that the other regions do not have strong effects, suggesting that the important drivers of functional diversity - primary production and anthropogenic disturbance - and are consistent across very different systems. From a basic ecology perspective that is an interesting, and suggests a consistent mechanism which you highlight (niche packing).

L526 - I think this limitations section is distracting and detracts from the study. As currently presented, it is simply a list of potential issues that could undermine results. I think it needs one of two approaches: 1) address each point and why the study still "holds." i.e., defend the ms against these critiques. This may be difficult and will take space. 2) Delete the paragraph entirely and distribute throughout ms. Most of these are methodological and could be addressed in a sentence following the introduction of the approaches. These are mostly known limitations, and a simple justification of the approach after introducing it would be sufficient IMO.

L634 - This conclusion only highlights one of two primary results, that occupancy-weighting detected response to NDVI. The other - human impacts - is noted in the title and should be highlighted here too. Moreover, these conclusions are more of a summary and could likely be cut entirely. My suggestion is to move the "limitations" throughout the ms (as noted above, #2) and use this conclusion to briefly highlight and then make suggestions for future work.

Fig 2 - Two points: 1) Should plot the intercept - in this case South America - so that all sites are represented. Right now it is just the betas. 2) Do the 3D FD plots include the extirpated species? As noted above, I think illustrating which species are lost and how that changes trait space would be interesting and help the reader understand how extinctions drive functional traits.

Author's Response to Decision Letter for (RSPB-2020-2098.R1)

See Appendix C.

Decision letter (RSPB-2020-2098.R2)

22-Jan-2021

Dear Mr Gorczynski

I am pleased to inform you that your manuscript entitled "Tropical mammal functional diversity increases with productivity but decreases with anthropogenic disturbance" has been accepted for publication in Proceedings B.

Your article has been estimated as being 9 pages long. Our Production Office will be able to confirm the exact length at proof stage.

Open Access

Paper charges

Sincerely,

Proceedings B

Appendix A

I just finished reviewing the manuscript entitled “Tropical mammal functional diversity increases with productivity and larger species pools and decreases with anthropogenic disturbance”. This is a nice study using functional diversity as a way for better testing classic biodiversity hypotheses. The authors used mammals as study system. Occurrence was taken from the largest standardized tropical forest camera trap monitoring system, the Tropical Ecology Assessment and Monitoring (TEAM) Network. This data was used to estimate mammal species occupancy from 15 tropical forest worldwide. Occupancy data was used as a proxy for species abundance, thus allowing the calculus of community weight mean (CWM) functional dispersion. The paper is well written and methods are appropriate. I have just a few suggestions which I describe below.

Functional diversity metrics

The authors calculated two metrics of functional diversity: functional richness and functional dispersion. For functional dispersion, they used occupancy values as a proxy for species abundance, and used these values for the calculus of CWM functional dispersion. I do not think there is a metric of CWM functional richness, thus I believe this is the reason why the authors did not included an estimate of CWM functional richness. This is, however, not mentioned in the methods section.

Calculating functional diversity metrics

Based on the R-code provided in the supplementary material, the authors entered a trait matrix containing all species in the global pool, including species within forest areas and the species pool. This is ideal, kudos! A single trait space must be used in order to preserve total inertia and distance between the same species occurring in different assemblages. I think this is an important detail worth mentioning in the methods section (e.g., see Villéger et al. (2017)). Still on the R-code, you used weights for traits (R-code line 70, $w = c(15,3,3,3,3,3,15,15,5,5,5,15)$). Inspecting traits and their associated weights, it seems to me that you tried to give equal weight (15) for each trait category (e.g., Diet has 5 categories so you give a weight of 3 for each category, summing a total of 15). The same was done for the other categorical traits. I am not familiar with this method and it begs me questions: 1) Why a total weight of 15 for each trait was used? 2) Is there any explanation that could be given to why traits should be weighted? 3) Is there any paper you can cite to support this choice? 4) What is the sensitivity of your results to this weight approach (would results change using different weights, let's say a total of 1 instead of 15)?

As seeing in the results of your analyses using weighted vs unweighted functional dispersion, weights can affect your results tremendously. Thus, an explanation for the use of weights for traits is required. Ideally, comparison of results with and without weights (or different weights) should be performed.

Finally, why you decided for not log-transform continuous trait variables? This can avoid inflating FD in communities containing mammals with large body size and large litter size.

TEAM data

Authors used data on ground-dwelling mammal species with an average body mass greater than 1 kg. Is there a reason why not including mammals with less than 1 kg? I think an obvious explanation would be that camera trap is not the best method for sampling small mammals. This caveat may be mentioned.

Analysis

The relative influence of predictor variables on CWM functional dispersion and functional richness was assessed based on linear regression models. All possible combination of variables were modeled and variable selection was done based on Aicc. The 2 best models were selected and their results reported

(see critic on this below). Considering that the objective of the study is to test the relative influence of multiple hypotheses for functional diversity, a better approach would be using a model averaging approach. Model averaging refers to the practice of using several models at once for making prediction. Model averaging produced parameter and error estimates not conditional on any single model (nor 'best-model') but instead derived from weighted averages of these values across multiple models (Burnham & Anderson, 2002). Model averaging can be calculated using the function "model.avg" from the R package MuMIn.

Results

The authors discuss results of the two best models based on AICc. I think discussing the two models makes the presentation of results unnecessarily long and confusing. I recommend presenting results of the best model only. Ideally, you should use model averaging as suggested above. Also, results were reported as non-standardized coefficients of linear models. It would be better using standardized coefficients because they allow comparison of model effects when the variables are measured in different units of measurement.

Minor comments:

Lines 80-85. Consider fragmenting this sentence for better reading.

Lines 217-218: Does this has any ecological meaning? Why is this presented here?

Table 1:

Diet: "Given a 1 if diet category composes >25% of average diet". Was it given 0 (zero) if <25% of average diet?

Activity Period: "Given a 1 if known to be consistently active during the time period". Was it given 0 (zero) if not active during the time period?

Citation

Burnham K. P. & Anderson, D. R. 2002. Model selection and multi-model inference: a practical information-theoretical approach. – Springer.

Villéger, S., Maire, E., & Leprieux, F. (2017). On the risks of using dendrograms to measure functional diversity and multidimensional spaces to measure phylogenetic diversity: A comment on Sobral et al. (2016). *Ecology Letters*, 20(4), 554–557.

Appendix B

Dear *Proceedings of the Royal Society B* Editorial Board,

We thank you for the opportunity to revise our manuscript “Tropical mammal functional diversity increases with productivity but decreases with anthropogenic disturbance” for consideration for publication in *Proceedings of the Royal Society B*. We have incorporated the editor and reviewer comments in this updated version of the manuscript as detailed in the following pages.

Understanding the drivers of biodiversity is important for advancing ecological theory and for conserving diversity in a changing world. Trait-based approaches have the potential to elucidate the mechanisms by which novel anthropogenic factors work in tandem with historical and environmental factors to influence global distributions of biological diversity. Tropical mammals are of particular interest given their unique ecological roles and vulnerability to anthropogenic threats, yet most previous global studies have relied on coarse gridded data extracted from geographic range maps. Here, we investigate the drivers of functional diversity in forest mammals using data from the largest global standardized tropical forest camera trap network, the Tropical Ecology Assessment and Monitoring (TEAM) Network. **We test how functional diversity varies in response to environmental, historical and anthropogenic factors for 15 protected tropical forests around the globe.**

We found that functional dispersion increased with productivity, while functional richness decreased with observed local extinctions. We also found significant differences in functional diversity between Madagascar and other regions, which suggests an important role of historical processes. Importantly, occupancy estimates from observational camera trap data revealed a novel, positive relationship between productivity and mammal functional diversity whereas previous range-map based studies reliant on presence only data have found the opposite or no relationship. Higher productivity may result in greater success and therefore higher abundances of species with unique functional traits.

Our findings suggest that energetic, historical and anthropogenic processes drive tropical mammal functional diversity. Given our novel trait-based global-scale analysis using high-resolution observational data, and the broad ecological and conservation implications of this study, we hope you will find our work suitable for *Proceedings of the Royal Society B*.

Thank you in advance for your time and consideration.

Best regards,

Daniel Gorczynski, PhD student,
on behalf of all coauthors

Editor

Comments to Author(s):

This manuscript has now been reviewed by two experts in the field. Their extensive comments are in general quite positive. Both referees see much value in the work, as do I. However, both have concerns about the appropriateness of the analyses and therefore strength of the results. In light of these, I am suggesting the ms be revised if the authors believe that after exploring alternative analyses the main results stand.

The two main issues in the analyses are around the use of regression. First, one referee believes that some analyses are highly leveraged, and suggests alternative approaches to explore the impact of this on the results. Second, there was a suggestion to take a more mixed model approach to address region/continental effects. Several other suggestions are made by both referees to improve the analyses.

The referees have also made numerous other suggestions aimed at improving clarity. One that I would like to emphasize is that the results section is too long, resulting in key messages being lost. The referees have made suggestions to address this. I would like to add that I think the discussion is also unnecessarily long.

Authors: Thank you for your time and consideration of this article. We have incorporated as much of the reviewers' feedback and suggestions as possible. Our previous results were largely robust to the recommended revised analyses, specifically the positive associations between productivity and functional dispersion, the negative association between local extinction and functional richness, and the significance of productivity based on occupancy-weighted but not unweighted functional dispersion. With the inclusion of biogeographical region as a covariate, we found significant differences in functional diversity among some regions and note that species pools no longer had a significant effect on functional richness.

For the revised analyses, we used a Bayesian regression framework as suggested by reviewer 2 and included biogeographical region as a fixed effect. We first tested biogeographical region as a random effect as suggested by reviewer 2, but the model containing the random effect consistently produced divergences suggesting poor convergence. Based on our sample size and potential presence of outliers, we elected not to use Reviewer 2 suggested that we implement leave-one-out cross validation because of our sample size and potential presence of outliers. In the leave-one-out cross validation method, when n is small, small differences in the predictive performance of similar models cannot be resolved, leading to uncertainty in results, even if there are no outliers in the data (Sivula, Magnusson & Vehtari 2020). This problem is compounded when outliers are present in the small sample (Bengio & Grandvalet 2004), which is likely the case in our data, as reviewer 2 points out. We therefore report a single global model for each functional diversity metric, which was recommended by reviewer 1. In addition, to account for possible outliers, we changed the distribution of the functional diversity models from Gaussian to Weibull distributions, as suggested by reviewer #2, because the response variables were left-skewed. We made additional improvements to the analysis and manuscript as requested by the reviewers, which we detail over the following pages, and we shortened both the results and discussion sections.

References

Bengio, Y. and Grandvalet, Y. No unbiased estimator of the variance of K-fold cross-validation. *Journal of machine learning research*, 5(Sep):1089–1105, 2004.

Sivula, T., Magnusson, M., & Vehtari, A. (2020). Uncertainty in Bayesian Leave-One-Out Cross-Validation Based Model Comparison. *arXiv preprint arXiv:2008.10296*.

Reviewer(s)' Comments to Author:

Referee: 1

Comments to the Author(s)

I just finished reviewing the manuscript entitled “Tropical mammal functional diversity increases with productivity and larger species pools and decreases with anthropogenic disturbance”. This is a nice study using functional diversity as a way for better testing classic biodiversity hypotheses. The authors used mammals as study system. Occurrence was taken from the largest standardized tropical forest camera trap monitoring system, the Tropical Ecology Assessment and Monitoring (TEAM) Network. This data was used to estimate mammal species occupancy from 15 tropical forest worldwide. Occupancy data was used as a proxy for species abundance, thus allowing the calculus of community weight mean (CWM) functional dispersion. The paper is well written and methods are appropriate. I have just a few suggestions which I describe below.

Authors: Thank you very much for your thoughtful and helpful comments, questions, and feedback.

Functional diversity metrics

The authors calculated two metrics of functional diversity: functional richness and functional dispersion. For functional dispersion, they used occupancy values as a proxy for species abundance, and used these values for the calculus of CWM functional dispersion. I do not think there is a metric of CWM functional richness, thus I believe this is the reason why the authors did not included an estimate of CWM functional richness. This is, however, not mentioned in the methods section.

Authors: We added the following sentence to the data section of the methods: “The functional richness metric is incapable of taking into account species abundances.”

Calculating functional diversity metrics

Based on the R-code provided in the supplementary material, the authors entered a trait matrix containing all species in the global pool, including species within forest areas and the species pool. This is ideal, kudos! A single trait space must be used in order to preserve total inertia and distance between the same species occurring in different assemblages. I think this is an important detail worth mentioning in the methods section (e.g., see Villéger et al. (2017)).

Authors: Thank you! This section in the second paragraph in the Methods-Data section now reads: “To preserve total inertia and distance between the same species occurring in different assemblages (e.g. 42), we calculated functional dispersion (34) and functional richness (43) for all sites and species pools in a single trait space (Fig. 1) using the dbFD function from the `FD` R package (44).”

Still on the R-code, you used weights for traits (R- code line 70, $w = c(15,3,3,3,3,3,15,15,5,5,5,15)$). Inspecting traits and their associated weights, it seems to me that you tried to give equal weight (15) for each trait category (e.g., Diet has 5 categories so you give a weight of 3 for each category, summing a total of 15). The same was done for the other categorical traits. I am not familiar with this method and it begs me questions: 1) Why a total weight of 15 for each trait was used?

Authors: We weighted traits with multiple categories so that the sum all of categories within a single trait was the same as the weight of a trait with a single category. We used a total weight of 15 for each trait since this allowed each trait and each category to be assigned a whole number. Thus R-code line 70, $w = c(15,3,3,3,3,3,15,15,5,5,5,15)$ corresponded to traits $c(\text{Body mass, Diet, Social, Substrate use, Activity period, Average liter size})$ with the following number of categories per trait $c(1, 5, 1, 1, 3, 1)$

2) Is there any explanation that could be given to why traits should be weighted?

Authors: Our goal was to weight the six traits evenly because we had no a priori expectations that certain traits were more important for functional diversity than others.

3) Is there any paper you can cite to support this choice?

Authors: The six traits we used have been considered equivalent in previous studies of mammal functional diversity (e.g., Oliveira et al. 2016 *Global Ecology and Biogeography*, Safi et al. 2011 *Philosophical Transactions of the Royal Society B*, Penone et al. 2016 *Proceedings of the Royal Society B*).

4) What is the sensitivity of your results to this weight approach (would results change using different weights, let's say a total of 1 instead of 15)?

Authors: The calculation was sensitive only to the ratio between the weights such that the functional diversity calculation using $w = c(15,3,3,3,3,3,15,15,5,5,5,15)$ was identical to $w = c(1,0.2,0.2,0.2,0.2,0.2,1,1,0.\bar{3},0.\bar{3},0.\bar{3},1)$ or $w = c(30,6,6,6,6,6,30,30,10,10,10,30)$. For example, here is our calculation of functional dispersion for each of the TEAM sites using the 15, 1 and 30 weight systems:

Site	Fdis.w15	Fdis.w1	Fdis.w30
BBS	0.241853	0.241853	0.241853
BCI	0.2695799	0.2695799	0.2695799
BIF	0.2702958	0.2702958	0.2702958
CAX	0.2572798	0.2572798	0.2572798
COU	0.2308579	0.2308579	0.2308579
CSN	0.2319487	0.2319487	0.2319487
KRP	0.2480309	0.2480309	0.2480309
NAK	0.2504147	0.2504147	0.2504147
NNN	0.2668109	0.2668109	0.2668109

PSH	0.2668664	0.2668664	0.2668664
RNF	0.2618553	0.2618553	0.2618553
UDZ	0.2670508	0.2670508	0.2670508
VB	0.1984659	0.1984659	0.1984659
YAN	0.263166	0.263166	0.263166
YAS	0.2564278	0.2564278	0.2564278

We have added the following sentence to the methods in the supplementary materials to improve clarity: “We had no a priori expectations that certain traits were more important for functional diversity than others. Similarly, previous studies of mammal functional diversity have given equal weight to the functional traits included in this study (Oliveira et al. 2016, Safi et al. 2011, Penone et al. 2016). We therefore weighted traits with multiple categories so that the sum all of categories within a single trait was the same as the weight of a trait with a single category.”

As seeing in the results of your analyses using weighted vs unweighted functional dispersion, weights can affect your results tremendously. Thus, an explanation for the use of weights for traits is required. Ideally, comparison of results with and without weights (or different weights) should be performed.

Authors: To assess whether weighting traits equally affected the directionality and significance of the predictors of functional diversity, we compared model results with and without equal weighting of traits as recommended and included these additional results in Figure S1. The functional richness results were unaffected by trait weighting, but the functional dispersion results without trait weighting had additional significant predictors. Specifically, local extinctions and the Madagascar bioregion also emerged as significant negative predictors of functional dispersion when weights to treat traits equally were excluded. Notably, these additional predictors were also significant predictors of functional richness. As described above, we retained the trait weights, which considered all traits equally, to adhere to the precedent established in previous studies of mammal functional diversity (Oliviera et al. 2016, Safi et al. 2011, Penone et al. 2016).

We included the following sentence in the Data section of the manuscript: “The traits were weighted such that all six traits contributed equally to the calculation.

“We also added the following in the results section: “To assess whether weighting traits equally in the dbFD function affected the directionality and significance of the predictors of functional diversity, we compared model results with and without equal weighting of traits. Functional richness was unaffected by trait weighting, but functional dispersion without trait weighting had additional significant predictors. Specifically, local extinctions and the Madagascar bioregion emerged as significant negative predictors of functional dispersion when weights to treat traits equally were excluded (Fig. S1). Notably, these additional predictors were also significant predictors of functional richness. We retained the weights, which considered all traits equally, to adhere to the precedent established in previous studies of mammal functional diversity (28, 35, 37).”

Below are coefficient plots showing model results with and without functional trait weights, which we included in the supplementary materials (Fig. S1):

Figure S1: Comparison of regression results for functional dispersion (a & b) and functional richness (c & d) when using weights in the FD function to treat all six functional traits equally (b & d) and when no weights were provided and the six functional traits were not treated equally (a & c). Functional richness was unaffected by trait weighting, but functional dispersion without trait weighting had additional significant predictors. Specifically, local extinctions and the Madagascar bioregion emerged as significant negative predictors of functional dispersion when weights to treat traits equally were excluded.

Finally, why you decided for not log-transform continuous trait variables? This can avoid inflating FD in communities containing mammals with large body size and large litter size.

Authors: We log-transformed both the body mass and the litter size trait per your suggestion. All results reported in the manuscript are based on these results, and the supplementary methods now reads: “Average body mass and average litter size were log-transformed because their distributions tend to be log-normal and to prevent inflation of functional diversity in sites with species with large body and litter sizes.”

TEAM data

Authors used data on ground-dwelling mammal species with an average body mass greater than 1 kg. Is there a reason why not including mammals with less than 1 kg? I think an obvious explanation would be that camera trap is not the best method for sampling small mammals. This caveat may be mentioned.

Author: We added the following statement to the supplementary methods: “We set a body mass threshold of one kilogram because other field techniques may be more effective for monitoring small mammal communities (Glen et al. 2013).”

Analysis

The relative influence of predictor variables on CWM functional dispersion and functional richness was assessed based on linear regression models. All possible combination of variables were modeled and variable selection was done based on AICc. The 2 best models were selected and their results reported (see critic on this below). Considering that the objective of the study is to test the relative influence of multiple hypotheses for functional diversity, a better approach would be using a model averaging approach. Model averaging refers to the practice of using several models at once for making prediction. Model averaging produced parameter and error estimates not conditional on any single model (nor ‘best-model’) but instead derived from weighted averages of these values across multiple models (Burnham & Anderson, 2002). Model averaging can be calculated using the function “model.avg” from the R package MuMIn.

Authors: Based on the comments of the other reviewer, we dropped the AIC model comparison of maximum likelihood linear regression models and instead used a single global Bayesian generalized linear regression, which had the recommended net effect of streamlining the results as this reviewer suggests here. Model averaging has recently been critiqued by leading quantitative ecologists (e.g., Grace & Irvine 2019 *Ecology*) and this was part of our rationale for adhering to the re-analysis recommendations from Reviewer #2.

Results

The authors discuss results of the two best models based on AICc. I think discussing the two models makes the presentation of results unnecessarily long and confusing. I recommend presenting results of the best model only. Ideally, you should use model averaging as suggested above. Also, results were reported as non-standardized coefficients of linear models. It would be better using standardized coefficients because they allow comparison of model effects when the variables are measured in different units of measurement.

Authors: We revised the manuscript to present the results of a single model only (see above). We also present model parameters as standardized beta coefficients with credible intervals. The analysis section of the methods now reads:

“None of the predictor variables were highly correlated ($r < 0.6$) and all continuous variables were scaled and centered to produce standardized beta coefficient estimates with a mean of zero and standard deviation of one.”

Minor comments:

Lines 80-85. Consider fragmenting this sentence for better reading.

Authors: The sentence in question now reads: “Two hypothesized mechanisms of the diversity-productivity relationship are based on resource availability and niche diversity. Resource availability in high productivity systems could benefit all species indiscriminately (23, 26) resulting in consistent functional diversity regardless of productivity. Alternatively, productivity could favor species with unique functional traits, resulting in increased functional diversity in high productivity systems (27, 25, 28).”

Lines 217-218: Does this has any ecological meaning? Why is this presented here?

Authors: We have revised the analysis to which this comment was referring, and no longer use Shapiro tests.

Table 1:

Diet: “Given a 1 if diet category composes >25% of average diet”. Was it given 0 (zero) if <25% of average diet?

Activity Period: “Given a 1 if known to be consistently active during the time period”. Was it given 0 (zero) if not active during the time period?

Authors: We have added these clarifying points to Table 1:

For diet: “Given a 1 if diet category composed >25% of average diet, 0 if diet category composed <25% of average diet”

For Activity period: “Given a 1 if known to be consistently active during the time period, 0 if known to not be consistently active during the time period”

Citation

Burnham K. P. & Anderson, D. R. 2002. Model selection and multi-model inference: a practical information-theoretical approach. – Springer.

Villéger, S., Maire, E., & Leprieur, F. (2017). On the risks of using dendrograms to measure functional diversity and multidimensional spaces to measure phylogenetic diversity: A comment on Sobral et al. (2016). *Ecology Letters*, 20(4), 554–557.

References

Estes, J. A., Terborgh, J., Brashares, J. S., Power, M. E., Berger, J., Bond, W. J., ... & Marquis, R. J. (2011). Trophic downgrading of planet Earth. *science*, 333(6040), 301-306.

Gaynor, K. M., Hojnowski, C. E., Carter, N. H., & Brashares, J. S. (2018). The influence of human disturbance on wildlife nocturnality. *Science*, 360(6394), 1232-1235.

Grace, J. B., & Irvine, K. M. (2020). Scientist’s guide to developing explanatory statistical models using causal analysis principles. *Ecology*, 101(4), e02962.

Oliveira, B. F., Machac, A., Costa, G. C., Brooks, T. M., Davidson, A. D., Rondinini, C., & Graham, C. H. (2016). Species and functional diversity accumulate differently in mammals. *Global Ecology and Biogeography*, 25(9), 1119-1130.

Penone, C., Weinstein, B. G., Graham, C. H., Brooks, T. M., Rondinini, C., Hedges, S. B., ... & Costa, G. C. (2016). Global mammal beta diversity shows parallel assemblage structure in similar but isolated environments. *Proceedings of the Royal Society B: Biological Sciences*, 283(1837), 20161028.

Safi, K., Cianciaruso, M. V., Loyola, R. D., Brito, D., Armour-Marshall, K., & Diniz-Filho, J. A. F. (2011). Understanding global patterns of mammalian functional and phylogenetic diversity. *Philosophical Transactions of the Royal Society B: Biological Sciences*, 366(1577), 2536-2544.

Referee: 2

Comments to the Author(s)

General comments: Gorczynski et al. tested 4 hypotheses – species pool hypothesis, spatial heterogeneity, primary production, and disturbance – for what drives functional diversity and dispersion. The present a clear question and approach coupled with a localized approach to estimating community composition via camera trap surveys. I think this is a nice, clean paper that tests multiple competing hypotheses and tells a compelling story. However, I have some general analytical concerns that could take significant work to overcome, and I have listed a number of specific comments to help improve clarity as well.

1. My biggest concern is the linear regressions. I ran the code and saw what appear to be a number of outliers as well as a potential lack of heteroscedasticity in residual variance. With only 15 data points I don't think this is particularly surprising but needs to be addressed more thoroughly. Classic approaches like transformations could help with residuals, but outliers with leverage will be harder to account for. I might recommend a Bayesian linear regression that can better assess model fits through leave-one-out cross validation that directly account for outliers. These models are easy to implement in brms and rstanarm (among others) with identical syntax to lm/lme4.

Authors: Thank you very much for these helpful insights, comments, and suggestions. We believe that the re-analysis is much stronger following the recommendations from this review. We have switched to a Bayesian regression model implemented in brms to deal with the issues described above. To account for possible outliers in the response variables, we changed the distribution of the models from Gaussian to Weibull distributions because the response variables were left-skewed. In leave-one-out cross validation, when n is small, small differences in the predictive performance of similar models cannot be resolved, leading to uncertainty in the results (Sivula, Magnusson & Vehtari 2020). This problem is compounded when outliers are present in the small sample (Bengio & Grandvalet 2004), which is likely the case in our data. Therefore, because of our small sample size and potential presence of outliers, we report a single global model (Vehtari & Ojanen 2012) for each functional diversity metric rather than implementing leave-one-out cross validation. Reviewer 1 recommended presenting results from a single model.

2. While limited in sample size, I think (and the authors note) that there are differences between regions in both response and predictors. This lends itself to a LMM approach with region/continent as a random effect to account for the underlying variation and hierarchical nature of clustered sampling/sites. While difficult to address with a standard LM/LMM approach, this would certainly be doable in a Bayesian framework. In general it looks like the data is good and the story is interesting, but the models are limiting.

Authors: We added biogeographical region to the analyses as requested. When we included biogeographical region as a random effect, the model consistently presented divergences despite our attempts to tweak it. We therefore included biogeographical region as a categorical fixed effect to ensure that the analysis sufficiently accounted for historical and geographical differences. We found that Madagascar had significantly lower functional richness than other regions while species pool functional richness was no longer a significant predictor. We hope that these adjustments to the analysis address your concerns.

Specific comments:

L37-43 – The abstract doesn't present any true results and I think it could use something here. What is the role/directionality of human presence, productivity, distance to settlement, etc?

Authors: Part of the abstract now reads: "Mammal community functional dispersion increased with primary productivity, while functional richness decreased with human-induced local extinctions and was significantly lower in Madagascar than other tropical regions. The significant positive relationship between functional dispersion and productivity was evident only when functional dispersion was weighted by species' occupancies. Thus, observational data from standardized monitoring can reveal the drivers of mammal communities in ways that are not readily apparent from range map-based studies. The positive association between occupancy-weighted functional dispersion of tropical forest mammal communities and primary productivity suggests that unique functional traits may be more beneficial in more productive ecosystems and may allow species to persist at higher abundances."

L80-85 – Move respectively to end of sentence. Also consider breaking this up. The mechanisms are important but get lost in these complex sentences.

Authors: The sentence in question now reads: "Two hypothesized mechanisms of the diversity-productivity relationship are based on resource availability and niche diversity. Resource availability in high productivity systems could benefit all species indiscriminately (23, 26) resulting in consistent functional diversity regardless of productivity. Alternatively, productivity could favor species with unique functional traits, resulting in increased functional diversity in high productivity systems (27, 25, 28)."

L89-97 – In a classical sense the role of disturbance in mediating diversity is similar to heterogeneity in that it creates opportunities for niche partitioning and mediates coexistence thereby increasing diversity. But human disturbances are extremely non-random and have known directionality. I think that needs some more unpacking here, as human disturbance likely does not have the same predicted effect as other forms of disturbance (e.g., fire).

Authors: This section now reads: "Natural disturbance, such as fire, can create novel habitats that support new functional traits (19) and prevent competitive exclusion (30). Anthropogenic disturbance, however, tends to negatively affect biodiversity through the selection of homogenized disturbance-adapted traits (31) resulting in local extinctions."

L130 – Ground-dwelling per se or simply terrestrial (non-arboreal) mammals?

Authors: We have changed the sentence to reflect the TEAM network's intent to monitor terrestrial mammals. The sentence now reads:

"All sites have been part of the Tropical Ecology Assessment and Monitoring (TEAM) Network and followed a standardized annual camera trapping protocol to monitor terrestrial (i.e. ground-dwelling) mammals (39)."

L134 – Clarify that this is citation 40 and not data from 40 sites – is that right?

Authors: The statement now reads: “data from Beaudrot et al. 2016 (40)”

L139 – Table 1 never cited, should go in this paragraph. A few additional details would also help: how was body mass calculated? Were activity and habitat in Table determined by cam data or lit review? Same with litter size. If these were means drawn from the citations that should be explicit. Also I think “habitat” is more like “arboreality.”

Authors: We have added the citation for Table 1 to the first paragraph of the Data section in the methods. The sentence now reads:

“We collected functional trait data for all study species through a search of published literature and databases (Table 1, Table S2, Table S3).”

We have also added the following statement to the caption of Table 1 “Values were drawn from published literature (see Table S3) and determined as the mean and/or consensus values from these published sources”. We changed the “Habitat” trait to “Substrate use”. We are concerned that the use of the term “arboreality” would introduce confusion since no truly arboreal species are included in the study. We only included scansorial species that have the ability to climb trees, but still spend the majority of their time on the ground.

L157-158 – Is this distance from centroid of MCP or mean distance from camera to settlement?

Authors: Given the decision to use a global model, we limited the number of predictor variables to avoid overfitting. We removed distance to settlement from the final analysis because there were already two other predictors of anthropogenic disturbance. As a result, the section in question has been omitted from the manuscript.

L161-163 – This local extinction is really more like the functional beta diversity which makes this circular no? As a predictor it measures the effect of changing functional richness on functional richness?

Authors: We included this predictor in our models to account for known changes in functional richness from recent species extinctions. The effect of species pool size or other predictor variables might have been influenced if we didn’t account for this known loss in functional richness. In addition, species extinctions do not necessarily change functional diversity because of functional redundancy in tropical systems (Gorzynski & Beaudrot 2020). We revised the text in the manuscript to clarify its inclusion:

“Finally, we included the impact of local species extinction on functional richness as an additional measure of anthropogenic disturbance and as an offset to control for losses in functional richness due to extirpations that have been documented at four TEAM sites over recent decades. Specifically Korup, Cameroon lost the leopard, golden cat and giant pangolin; Barro Colorado Island and Soberania National Park, Panama lost the white-lipped peccary, giant anteater and likely the jaguar; Bwindi Impenetrable Forest, Uganda lost the buffalo, leopard, and giant forest hog; Nam Kading, Laos lost both the tiger and leopard (38). We note that local extinctions may not necessarily impact functional diversity because of the functional redundancy that occurs in tropical forest mammal communities (48). We quantified local extinction for each TEAM site as the difference in functional richness between the functional richness of the current realized community and the functional richness of current realized community as well as the recently extirpated species.”

L166-173 – Are the sites independent? It seems to me that you have 3 distinct regions – South America, Africa, and Southeast Asia – that could also have their own structure, especially with respect to predictors like species pools, NPP, and disturbance. I think a mixed effects model would be a better application here as it would not only account for this underlying variation but would also tell you if specific regions diverge from the overall pattern presented. This would provide important information for regional management and conservation that is currently not represented.

Authors: Thank you for the suggestion. We attempted to include biogeographic region as a random effect but were unable to prevent divergences in the model. We therefore included biogeographical region as a categorical fixed effect to account for regional differences and potential lack of site independence.

The methods section of the manuscript now includes the following text, “The global model consisted of all potential predictor variables, plus a categorical fixed effect for biogeographical regions (Neotropics, Africa, Asia, Madagascar) to account for region-specific patterns in functional diversity. We used the `brm` function from the `brms` (49) package in R to fit the models.”

The results section of the manuscript now includes the following text: “There was also a significant negative effect for Madagascar ($\beta = -2.61$, $\text{Err} = 0.48$), indicating that functional richness was significantly lower at the Malagasy site than sites in other regions.”

L174-179 – This needs more explanation. The intro emphasizes the importance of local occupancy for calculating site level FD/FR, but its buried in the methods and this is the chance to show that utility. I’m also not entirely sure what was done here. Occupancy was used to weight FD/FR (how? Basic description would better than a citation in my opinion), and then calculated unweighted using camera data and range maps (classical approach)? A few extra lines outlining this would provide more clarity and help future researchers recreate your approach, which I would guess is a primary objective here.... I now see the full materials and methods in S11, but it’s never cited. A condensed methods is fine, but the key points need to be clearer and the full methods need to be cited so anyone can go check them.

Authors: We have added the following explanations to the Data section and the Analysis section. These sections now read:

Data Section: “To preserve total inertia and distance between the same species occurring in different assemblages (e.g. 42), we calculated functional dispersion (34) and functional richness (43) for all sites and species pools in a single trait space (Fig. 1) using the `dbFD` function from the `FD` R package (44). The traits were weighted such that all six traits contributed equally to the calculation. Functional dispersion measures the distribution of species in trait space or how similar a community is in terms of its functional traits. Functional dispersion can either be unweighted, with all species accounted for equally, or weighted by species abundances, with distance from the community centroid to abundant species in trait space contributing more to the metric. Functional richness measures the volume of the convex hull encompassing all species in a community in trait space. The functional richness metric is incapable of taking into account species abundances.”

Analysis section: “To assess the importance of occupancy in functional dispersion, we also ran regression models for two additional unweighted functional dispersion calculations. Unlike occupancy-weighted functional dispersion, the additional calculations treated all species equally and did not incorporate variation in species abundances. As a result, they likely omit important ecological information about the relative abundances of functional traits. We calculated unweighted functional dispersion using 1) presence data of the observed community from camera trap data, and separately using 2) presence data extracted from IUCN range maps. See Appendix S10 for full materials and methods.”

L182-194 – I’m generally not a fan of tables, but this whole paragraph is summary stats that could be summarized more efficiently in a table and would save a lot of text for more important info (e.g., additional methods context).

Authors: We removed the paragraph and added a supplementary table (Table S4) with the minimum, median and maximum for each of the protected area-level variables.

L196 – I would use beta/ β instead of est

Authors: We changed “est” to β throughout the results section.

L195-208 – I ran the code and checked the models because I was skeptical about only 15 data points detecting significance of multiple predictors. Between the Shapiro-test and the QQ plot its clear, as noted, that the residuals are normally distributed, but a fitted vs residual plot shows a pretty clear lack of heteroscedasticity (larger fitted values have much more variance than smaller). It also looks like there are a number of outliers for most of the predictors, and I would be worried about the leverage these have given the small sample size. I think some transformations are likely in order, or a Bayesian linear regression that can handle complex models with limited data (these are now very easy to implement in packages like brms and rstanarm that have the same syntax as lm/lme4). I’d also note that neither Fdis nor Frich look like they are normal distributions, so a transformation or a generalized lm with an alternative distribution might help with this.

Authors: Thank you very much for this helpful comment and the suggestions for implementation. Based on your concerns about limited data, heteroscedasticity, and outliers we re-analyzed the data in a Bayesian framework using the brm function from the brms and rstanarm packages, which, as you mention, is able to handle complex models with limited data. We report a single global model with all predictor variables, which we interpret based on credible intervals. The brms package is a very useful tool for this analysis. In addition, we used a Weibull distribution, which more appropriately fit the Fdis and Frich response variables than a Gaussian distribution. We edited the following section clarifying this in the Supplementary methods:

“Given the left-skew in the distributions of both functional diversity metrics, we specified a Weibull distribution for both models. The Weibull distribution is an unbounded continuous probability distribution that can accommodate skews and long tails in the data. The global model consisted of all potential predictor variables, plus a categorical fixed effect for biogeographical region (Neotropics, Africa, Asia, Madagascar) to control for region-specific patterns in functional diversity. The Neotropics was the category to which all other bioregions were compared. We used the brm function from the brms (Bürkner 2017) package in R to fit the models.”

L251-254 – This is interesting because landcover.div vs. Frich in the data looks like an exponential pattern. It’s also possible that heterogeneity in the form of landscape shape, not just composition, could drive functional richness.

Authors: We substituted transformed versions of the landcover.div variable in the model but did not find changes in effect size, direction or significance for the predictor variables. See below for a comparison of the model results with log-transformed versus non-transformed landcover. We agree that there may be other ways of measuring habitat heterogeneity that are relevant to mammal functional diversity and added the following statement to the discussion section: “It is also possible that the spatial arrangement of habitat types could affect functional diversity.”

Below are coefficient plots showing model results with and without log transformed landcover diversity

With transformed Landcover diversity

Without transformed Landcover diversity

L268-269 – This is very interesting, and again I think I mixed-effects structure would help pull out this effect and quantify it. I would also be interested in the interaction between species pools and human disturbance, though you might be data limited in terms of sites.

Authors: Please see above for further details on inclusion of bioregion as a fixed effect. We ran the models with interaction effects for species pool and human density as well as species pool and local extinctions. While the models converged, neither interaction effect was a significant predictor of either functional diversity metric. We therefore did not include interaction effects in the revised manuscript.

L293-297 – Which traits are driving this pattern? I think this is true, but as with most functional traits at this coarse resolution (binary indicators of list history traits) it's hard to tie an actual function or niche to them. It would be nice to show a mechanism. You talk about this limitation below, but being able to tie some of these potential consequences to species.

Authors: Unfortunately, methods of this kind are not currently available. The FD framework does not allow users to assess which traits are the ones driving differences in functional diversity between sites. Obtaining these results would require major additions to functions in the FD package. Studies will sometimes overcome this issue by using single trait analyses over multi-trait analyses to infer causal links

(Butterfield & Suding 2013), but we used a multi-trait analysis because of the multi-dimensional nature of tropical mammal niches. Although comparisons like these would be very interesting to pursue in the future, developing a new measure of functional structure would be beyond the scope of this study. We added the following to the limitations section to address this issue:

“The multi-trait functional diversity measure we used cannot infer mechanistic links between individual traits and environmental drivers.”

References:

Bengio, Y. and Grandvalet, Y. No unbiased estimator of the variance of K-fold cross-validation. *Journal of machine learning research*, 5(Sep):1089–1105, 2004.

Butterfield, B. J., & Suding, K. N. (2013). Single- trait functional indices outperform multi- trait indices in linking environmental gradients and ecosystem services in a complex landscape. *Journal of Ecology*, 101(1), 9-17.

Sivula, T., Magnusson, M., & Vehtari, A. (2020). Uncertainty in Bayesian Leave-One-Out Cross-Validation Based Model Comparison. *arXiv preprint arXiv:2008.10296*.

Vehtari, A., & Ojanen, J. (2012). A survey of Bayesian predictive methods for model assessment, selection and comparison. *Statistics Surveys*, 6, 142-228.

Appendix C

Dear Proceedings of the Royal Society B Editorial Board,

We thank you for the opportunity to revise our manuscript “Tropical mammal functional diversity increases with productivity but decreases with anthropogenic disturbance” for publication in Proceedings of the Royal Society B. We have incorporated the suggested revisions in this updated version of the manuscript.

Understanding the drivers of biodiversity is important for advancing ecological theory and for conserving diversity in a changing world. Trait-based approaches have the potential to elucidate the mechanisms by which novel anthropogenic factors work in tandem with historical and environmental factors to influence global distributions of biological diversity. Tropical mammals are of particular interest given their unique ecological roles and vulnerability to anthropogenic threats, yet most previous global studies have relied on coarse gridded data extracted from geographic range maps. Here, we investigate the drivers of functional diversity in forest mammals using data from the largest global standardized tropical forest camera trap network, the Tropical Ecology Assessment and Monitoring (TEAM) Network. We test how functional diversity varies in response to environmental, historical and anthropogenic factors for 15 protected tropical forests around the globe.

We found that functional dispersion increased with productivity, while functional richness decreased with observed local extinctions. We also found significant differences in functional diversity between Madagascar and other regions, which suggests an important role of historical processes. Importantly, occupancy estimates from observational camera trap data revealed a novel, positive relationship between productivity and mammal functional diversity whereas previous range-map based studies reliant on presence only data have found the opposite or no relationship. Higher productivity may result in greater success and therefore higher abundances of species with unique functional traits.

Our findings suggest that energetic, historical and anthropogenic processes drive tropical mammal functional diversity. Given our novel trait-based global-scale analysis using high-resolution observational data, and the broad ecological and conservation implications of this study, we hope you will find our work suitable for Proceedings of the Royal Society B.

Thank you in advance for your time and consideration.

Best regards,

Daniel Gorczynski, PhD student,
on behalf of all coauthors

Reviewer(s)' Comments to Author:

Referee: 2

Comments to the Author(s)

RSPB-2020-2098.R1 – “Tropical mammal functional diversity increases with productivity but decreases with anthropogenic disturbance”

Gorcynzski et al. have done a nice job with reviewer suggestions and I feel that the ms is much improved. In particular, I believe the transition to a Bayesian regression approach is much more appropriate, and I fully agree that a single, global model testing the relative effects of each competing hypothesis is the way to go here. I also like the added emphasis on occupancy-weighted measurements compared to range-based approaches, as it shows the novelty and added utility of the approach presented herein. My remaining comments are largely editorial and aimed at clarifying the ms. Nicely done!

Specific Comments (line numbers from tracked changes ms in Response):

L101-102 – “increasing the energetic capacity of an ecosystem and allowing greater energy allocation to individual organisms” - I think this would read better as “presumably due to increased energetic capacity...” As mentioned, there are multiple mechanisms in play and the diversity-productivity debate is not especially settled.

Authors: The sentence has been changed to reflect the uncertain nature of the productivity mechanism. It now reads:

“Thirdly, primary productivity is an important correlate of plant and animal biodiversity across large spatial scales (22, 23, 24), potentially due to increased energetic capacity of the ecosystem and greater energy allocation to individual organisms (25).”

L129 – affect

Authors: The word “affects” has been changed to “affect”.

L131-133 – This sentence reads oddly. I think it should be “the relative strength of anthropogenic disturbance on tropical mammal community assembly” or “the relative importance of anthropogenic disturbance in tropical mammal community assembly.” Also, I think you need to clarify what the strength is relative to. Other regions? Other drivers? Given that you are comparing 4 hypotheses, I think it’s the latter, but the subject of the sentence (gradient across sites) suggests the former.

Authors: The sentence in question now reads:

“Comparing protected areas exposed to different degrees of anthropogenic disturbance may reveal the relative importance of anthropogenic disturbance in tropical mammal community assembly in comparison to other drivers.”

L139 – Temperate communities

Authors: The phrase “temperate species” has been changed to “temperate communities”

L204-205 – I think some discussion of why these traits matter is needed. Are they response or effect traits? The introduction mentions the importance of mammals and their trait variation for ecosystems, this would be the spot to unpack that (or in the table). Too many macroecological studies lose sight of the actual functions associated with functional traits. I don’t think that’s the case here as most traits do map to, or scale with, specific functions/services...but some justification of why these traits are representative of functional diversity and why we should care about changes to them would be good.

Authors: We have added the following section to the methods detailing the ecological importance of the six traits we selected:

“We chose six functional traits to calculate functional diversity: 1) average body mass, 2) diet composition, 3) social or asocial behavior, 4) scansorial or entirely terrestrial substrate use, 5) activity period, and 6) average litter size. These functional traits relate to both the response of the species to environmental conditions (response traits) and to the role of the species in an ecosystem (effect traits; 44). Body mass affects the quality and quantity of resources necessary for survival, and also approximates the impact that the species may have on the ecosystem in terms of spatial range use, nutrient dispersal and trophic regulation. Diet composition characterizes the resources a species requires, but also identifies other taxa in the ecosystem with which a species potentially interacts. Social group size can alter species’ allocation of time to different behaviors including foraging, predator avoidance, and care of offspring, and also indicates how the species’ impact will be distributed in space. Substrate use characterizes where a species can obtain resources and where a species will directly interact with an ecosystem. Activity period characterizes when a species obtains resources and interacts with an ecosystem. Finally, litter size characterizes the life history strategy of a species and indicates how the ecological impact of a species can vary temporally based on population dynamics.”

L276-288 – I think this is a much-improved framework to test competing hypotheses with limited data. In addition to manual trace examinations, you should also present (supplementary) posterior predictive checks of your errors and distributions to show that your models accurately reflect the data – see here (<https://mc-stan.org/bayesplot/reference/PPC-overview.html#ppc-plotting-functions>). As is, it is still not clear whether the model fits are good, especially since no model selection was used to assess fits (though I agree with your decision to use a single, global model).

Authors: We have added the following sentence to the methods section:

“We visually assessed model trace plots (Fig. S1), used Rhat criteria (< 1.05) to assess convergence and performed graphical posterior predictive checks on model distributions and error (Fig. S2).”

And we added the following figures to the supplementary materials:

Figure S1: Trace plots from functional dispersion and functional richness model outputs

Figure S2: Graphical posterior predictive checks of the distribution (a) and error (b) of the functional dispersion model, and the distribution (c) and error (d) of the functional richness model.

L324 – One of the best things about a Bayesian approach is that it frees you from the confines of statistical significance in the classical sense. I think it's fine to use credible intervals not overlapping zero as a measure of significance, but the Bayesian framework also allows you to assess the relative strengths of your multiple competing hypotheses even if they are not significant per se...e.g., extinction and site (Madagascar) had "significant" effects on richness, but even when accounting for those the effect of species pools was still positive and substantial, just not as strong. That seems notable, and given the relatively small sample size used here it's not unreasonable to think you simply lacked the power or variation to detect a stronger effect.

Authors: We added the following section to the discussion, acknowledging the substantial, though not significant, effect of species pools on functional richness:

"Species pools were also not a statistically significant driver of either functional diversity metric, but did have a positive effect on functional richness. The sample size in this study may have been too small to detect a significant effect of species pools. Nevertheless, the non-significant positive effect suggests that larger species pool functional richness may facilitate larger realized community functional richness if it increases the likelihood that a pool contains species with functional traits that can occupy all available niches in a community (1)."

L509 – Change to "Our results indicate that recent, human-induced local extinctions determined mammal functional richness in the tropical forest protected areas we surveyed." ... improve concision

Authors: The sentence in question has been changed to:

"Our results indicate that recent, human-induced local extinctions in part determined mammal functional richness in the tropical forest protected areas we surveyed."

L519-521 – The loss of predators is indeed notable, but not surprising (see/cite Estes et al. 2011, Ripple et al. 2014, Science). Carnivores have disproportionate effects on ecosystem processes, and the loss of these species and their associated traits likely has cascading impacts on the rest of the system. Your results emphasize that point.

Authors: We revised the following sentence in the discussion regarding previous research on the disproportionate effect of carnivores on ecosystem processes and trophic cascades:

"Notably, sites that experienced local extinctions all lost their top predator. Previous research has shown that top predators are often extirpated first from systems under anthropogenic pressure, and that their loss often has disproportionate effects on the ecosystem due to cascading impacts on lower trophic levels and ecosystem services (55, 56). Korup, which lost its top predator as well as its only other obligate carnivore (African golden cat) and a unique specialist species (giant pangolin), had the greatest decline in functional richness. Our study reinforces that the loss

of top predators and specialists will likely result in disproportionate losses of functional diversity as anthropogenic influences continue to alter tropical mammal communities (32, 33).”

An interesting way to show this would be highlighting extirpated species in Fig 1 to illustrate how these species impact trait space.

Authors: Thank you for the suggestion. We have made changes to figure 2 to include data points for extirpated species at sites that have experienced local extinctions. We did not make changes in Figure 1 because Figure 1 displays a single trait space for all species at all sites; species extirpated at one site were present at other sites and so their removal from the figure would have been misleading. We added a sentence in the caption of Figure 2 to clarify the role of species extinction in changing trait space:

“Red circles (in d) indicate species that were extirpated in recent decades and were not part of the functional richness calculation. The species positioned exterior to the convex hull demonstrate how their extirpation caused a decrease in community trait space.”

L521-524 – This is the only real mention of geographic region. As you note, Madagascar has a very unique phylogeographic history which likely drives the pattern. It is also the only island, has only 1 site, and is by far the smallest of the regions. All of these also likely impact functional diversity (and probably should be noted). More notably though, is that the other regions do not have strong effects, suggesting that the important drivers of functional diversity – primary production and anthropogenic disturbance – and are consistent across very different systems. From a basic ecology perspective that is an interesting, and suggests a consistent mechanism which you highlight (niche packing).

Authors: We added the following two sentences to the discussion to address these points:

“The fact that the single Malagasy TEAM site, Ranomafana, had substantially lower species richness (58) and functional diversity than TEAM sites in other biogeographical regions may also be partly attributable to the small size and insular nature of Madagascar.”

“The lack of significant effects for Africa, Asia and the Neotropics suggests that the effects of productivity and local extinction were similarly influential across very different systems and therefore that consistent mechanisms may be operating across diverse tropical regions.”

L526 – I think this limitations section is distracting and detracts from the study. As currently presented, it is simply a list of potential issues that could undermine results. I think it needs one of two approaches: 1) address each point and why the study still “holds.” i.e., defend the ms against these critiques. This may be difficult and will take space. 2) Delete the paragraph entirely and distribute throughout ms. Most of these are methodological and could be addressed in a sentence following the introduction of the approaches. These are mostly known limitations, and a simple justification of the approach after introducing it would be sufficient IMO.

Authors: The limitations section has been deleted and distributed as requested.

An excerpt in the study sites section in the Methods now reads:

“We considered each site to be a community and were not able to account for variation within protected areas.”

These excerpts in the data section in the Methods now read:

“Occupancy is not equivalent to abundance (41) but was used because of its practical advantages for infrequently detected tropical mammal species (42).”

“We did not consider strictly arboreal mammals, semi-aquatic mammals, volant mammals, or mammals less than 1 kilogram, which could limit comparability to global studies that include additional mammals.”

“As with any functional trait study, our results are in large part dependent upon the traits considered and the diversity encompassed by functional metrics can change with the inclusion or exclusion of specific traits (44). Traits used in this study were gathered at the species-level and multiple traits were reduced to binary variables, which may reduce the breath of trait variation to some extent.”

“...and neither metric can infer mechanistic links between individual traits and environmental drivers.”

L634 – This conclusion only highlights one of two primary results, that occupancy-weighting detected response to NDVI. The other – human impacts – is noted in the title and should be highlighted here too. Moreover, these conclusions are more of a summary and could likely be cut entirely. My suggestion is to move the “limitations” throughout the ms (as noted above, #2) and use this conclusion to briefly highlight and then make suggestions for future work.

Authors: We revised the conclusion as suggested. The section now reads:

“This study suggests that higher productivity in tropical most forests may allow greater success of mammal species with diverse niche and support more functionally diverse communities as a result. At the same time, anthropogenically-induced extirpations may be causing significant declines in functional diversity in these communities, despite their high functional redundancy. Based on these findings, we strongly encourage the continuation and expansion of standardized monitoring studies to allow a greater understanding of community assembly and disassembly in the Anthropocene.”

Fig 2 – Two points: 1) Should plot the intercept – in this case South America – so that all sites are represented. Right now it is just the betas. 2) Do the 3D FD plots include the extirpated species? As noted above, I think illustrating which species are lost and how that changes trait space would be interesting and help the reader understand how extinctions drive functional traits.

Authors: 1) We added the intercept to the effect plots in Figure 2 as requested and added the following sentence to the caption to clarify the definition of the intercept:

“The intercept effects, which are outputs of the Weibull models, represent the log-transformed functional diversity measurement of the baseline biogeographic region (i.e. the Neotropics) when all other covariates are held at zero.”

2) We added extirpated species to the trait plots as red dots (described in depth above) and added the following sentences to the caption clarifying this argument:

“Red circles (in d) indicate species that were extirpated in recent decades and were not part of the functional richness calculation. The species positioned exterior to the convex hull demonstrate how their extirpation caused a decrease in community trait space.”